# Enhancement of snow albedo reduction and radiative forcing due to coated black carbon in snow

Wei Pu[1], Tenglong Shi[1], Jiecan Cui[1], Yang Chen[1], Yue Zhou[1], Xin Wang[1,2]

[1]Key Laboratory for Semi-Arid Climate Change of the Ministry of Education, College of Atmospheric Sciences, Lanzhou University, Lanzhou 730000, China

[2]Institute of Surface-Earth System Science, Tianjin University, Tianjin 300072, China

Corresponding author: Xin Wang (wxin@lzu.edu.cn)

**Abstract** When black carbon (BC) is mixed internally with other atmospheric particles, the BC light absorption effect is enhanced. This study explicitly resolved the optical properties of coated BC in snow based on the core/shell Mie theory and the snow, ice, and aerosol radiative model (SNICAR). Our results indicated that the BC coating effect enhances the reduction in snow albedo by a factor ranging from 1.1–1.8 for a nonabsorbing shell and 1.1–1.3 for an absorbing shell, depending on the BC concentration, snow grain radius, and core/shell ratio. We developed parameterizations of the BC coating effect for application to climate models, which provides a convenient way to accurately estimate the climate impact of BC in snow. Finally, based on a comprehensive set of in situ measurements across the Northern Hemisphere, we determined that the contribution of the BC coating effect to snow light absorption exceeds that of dust over northern China. Notably, high enhancements of snow albedo reduction due to the BC coating effect were found in the Arctic and Tibetan Plateau, suggesting a greater contribution of BC to the retreat of Arctic sea ice and Tibetan glaciers.

## 1 Introduction

Snow is the most reflective natural substance on the surface of Earth and covers more than 30% of the global land area (Cohen and Rind, 1991). Snow albedo feedback is considered one of the major energy balance factors of the climate system. Previous observations have revealed that light-absorbing particles (LAPs; e.g., black carbon (BC), organic carbon (OC), and mineral dust) within snow may reduce snow albedo and enhance the absorption of solar radiation (Hadley and Kirchstetter, 2012). As a result, LAPs play a major role in the alteration of snow morphology and snowmelt processes and therefore yield important effects on local hydrological cycles and global climate (Qian et al., 2009).

Given the importance of the climate feedback caused by LAPs in snow, studies have developed snow radiative models and sought to improve our understanding of the influence of LAP-contaminated snow on climate. For example, Warren and Wiscombe (1980) developed a radiative forcing model based on the Mie theory and the δ-Eddington approximation and reported that snow albedo at visible wavelengths could be reduced by 5%–15% with 1000 ng g$^{-1}$ BC in snow. Flanner et al. (2007) established a more comprehensive snow albedo model (the snow, ice, and aerosol radiation model; SNICAR) for a multilayer snowpack based on the two-stream radiative transfer solution. In addition to BC, the SNICAR model also considers the potential effects of dust particles and volcanic ash on snow albedo. Recently, studies have indicated that the mixing state of BC and snow could effectively alter snow albedo (Liou et al., 2011,

2014; Flanner et al., 2012; Liu et al., 2012; He et al., 2017, 2018a, b, c). Moreover, the snow grain shape exerts an important influence on snow albedo (Kokhanovsky and Zege, 2004). Nonspherical snow grains attain a lower albedo reduction than that due to spherical snow grains (He et al., 2018c; Dang et al., 2016).

Although efforts have been made to optimize snow albedo models, current models still suffer from major limitations. Studies have demonstrated that when BC in the atmosphere is coated with other aerosols, this greatly enhances light absorption via a lensing effect over uncoated BC, as investigated via model simulations (e.g., Jacobson 2001; Matsui et al., 2018) and experimental measurements (e.g., Cappa et al., 2012; Peng et al., 2016). Moreover, coated BC has been observed to persist for only a few hours after emission in certain regions (Moteki et al., 2007; Moffet and Prather, 2009). Global aerosol models that simulate microphysical processes have revealed that most BC is mixed with other particles within 1–5 days (Jacobson, 2001) at all altitudes (Aquila et al., 2011). However, it remains uncertain whether coated BC occurs in real snowpacks because the coating materials (e.g., salts and OC) other than BC may dissolve during wet deposition. A recent study observing the individual particle structure and mixing states between glaciers–snowpacks and the atmosphere based on field measurements and laboratory transmission electron microscopy (TEM) and energy dispersive X-ray spectrometry (EDX) instrument analysis (Dong et al., 2018) has provided an answer. It was found that salt-coated BC was still observed in real snowpacks despite its lower proportion than that in the atmosphere due to the

dissolution effect during snow precipitation. Regarding OC, the above study did not observe reduced OC components in LAPs. More notably, it was also determined that the proportion of coated BC was even higher in snowpacks than that in the atmosphere. All of the above observation results demonstrate that coated BC particles occur in real snowpacks and are even more common than those occurring in the atmosphere. Hence, the climate impacts of BC must be evaluated within the context of the BC coating effect on light absorption enhancement.

Although the BC coating effect on light absorption enhancement in the atmosphere has been broadly acknowledged, little research has been carried out on snow albedo. Flanner et al. (2007) developed the first radiative transfer model to investigate the coating effect on snow albedo, thereby employing sulfate as BC particle coating material with a constant absorption enhancement factor of ~1.5. Subsequently, Wang et al. (2017) considered a similar constant light absorption enhancement factor in their spectral albedo model for dirty snow (SAMDS). However, the above factor varies with the optical properties of different coating materials, core/shell ratio, wavelength, and other parameters in real environments (Lack and Cappa, 2010; Liu et al., 2017). For example, Liu et al. (2017) reported that the core/shell ratio notably controls light absorption enhancement. You et al. (2016) suggested that light absorption enhancement is highly correlated with visible or near-infrared (NIR) wavelengths and coating material. Furthermore, a core/shell Mie theory-based simulation study (Lack and Cappa, 2010) found that the attained light absorption enhancement was smaller for mildly

absorbing coatings (e.g., OC) than that attained for nonabsorbing coatings (e.g., sulfate).

Hence, the use of a constant enhancement factor may result in biased simulation

estimates, which prevents us from obtaining a better understanding of the hydrological

and climate impacts of BC in snow.

In this study, we apply the core/shell Mie theory to calculate the optical properties

of coated BC considering both mildly absorbing OC and nonabsorbing sulfate and

incorporate these results into the SNICAR model to evaluate the influence on snow

albedo. Parameterizations of the BC coating effect are then developed for application

in other snow albedo and climate models. Finally, we estimate the enhancement of snow

albedo reduction and the associated radiative forcing due to the BC coating effect across

the Northern Hemisphere by combining model simulations with in situ observations of

the BC and OC concentrations in snow.

**2 Methods**

**2.1 Modeling**

**2.1.1 Optical parameter calculations for snow coated in BC**

Figure 1a and 1c show schematics of light absorption by externally and internally

mixed particles (EMPs and IMPs, respectively). EMPs are particles not coated in BC

mixed with other particles, while IMPs include BC, which is assumed to be the core

material coating particles and acts as a shell (Kahnert et al., 2012). Regarding the

nonabsorbing shell, the overall light absorption includes contributions of the BC core

and absorption enhancement due to the lensing effect, while regarding the absorbing

shell, the shell itself also contributes to light absorption. The lensing effect indicates

that when BC is coated with a nonabsorbing shell (or an absorbing shell), the shell acts

as a lens and focuses more photons onto the core than would reach it otherwise so that

the light absorption effect of the BC core is enhanced (Bond et al., 2006).

To evaluate the BC coating effect on snow albedo, it is necessary to determine the

optical parameters of coated BC. The refractive index (RI) of BC was assumed to be

1.95–0.79i following Lack and Cappa (2010), which is consistent with the original

SNICAR model (Flanner et al., 2007). Two types of particle shells (nonabsorbing and

absorbing) were considered. The nonabsorbing shell was represented with sulfate, and

its RI was set to be $1.55–10^{-6}i$ following the atmospheric study of Bond et al. (2006).

The absorbing shell was represented with OC, which is a major light-absorbing particle

in snow (Wang et al., 2013). The RI of OC varies with the wavelength. Here, a fixed

mass absorption coefficient (MAC) for OC of 0.3 $m^2$ $g^{-1}$ at 550 nm, a real RI value of

1.55, and a particle diameter of 200 nm were assumed, following the observations of

Yang et al. (2009) and the study of Lack and Cappa (2010). The uncertainty in snow

albedo considering the BC coating effect due to the OC MAC will be discussed in

Section 3.4. Based on the Mie theory, an imaginary RI value of $-1.36 \times 10^{-2}i$ at 550 nm

was calculated. Subsequently, wavelength-dependent imaginary RI values (Figure S1)

were derived according to an absorption angstrom exponent (AAE) of −6 (Sun et al.,

2007).

In regard to a core/shell-structured particle, the core and shell diameters refer to

the BC core diameter and the whole-particle diameter, respectively. The BC diameter
usually ranges from ~50–120 nm in the atmosphere (Corbin et al., 2018) and are
typically larger by ~20 nm in snow due to the removal process via wet deposition
(Schwarz et al., 2013). Therefore, we assumed that the BC diameter in snow was 100
nm with a fixed monodisperse size distribution. The uncertainty in snow albedo of the
BC coating effect due to the BC size distribution will be described in Section 3.4. The
shell diameter was assumed to range from 110 to 300 nm based on Bond et al. (2006).
The above core and shell diameters, RI, and wavelength were then applied in a Mie
model to derive the optical parameters of core/shell particles, including the single
scatter albedo (SSA), asymmetry factor ($g$), and extinction cross-section ($Q_{ext}$). The
mass extinction coefficient (MEC) of core/shell particles was calculated based on $Q_{ext}$
and the density, given as 1.8 g cm$^{-3}$ for BC (Bond et al., 2006), 1.2 g cm$^{-3}$ for sulfate,
and 1.2 g cm$^{-3}$ for OC (Turpin and Lim, 2001).
**2.1.2 Snow albedo calculations**
We simulated snow albedo with the SNICAR model (Flanner et al., 2007), which
calculates the radiative transfer in a snowpack based on the theory of Warren and
Wiscombe (1980) and a two-stream multilayer radiative approximation (Toon et al.,
1989). Here, we summarize only the model features in SNICAR that are crucial to our
study. The SNICAR model allows for a vertical multilayer distribution of snow
properties, LAPs, and heating throughout the snowpack column. The input optical
parameters (MEC, SSA, and $g$) of snow grains and BC were calculated offline with the
Mie theory. SNICAR provides snow albedo changes due to uncoated and sulfate-coated
BC particles, in addition to dust particles and volcanic ash (for further details, please
refer to Flanner et al., 2007).

4        In this study, we assumed a homogeneous semi-infinite snowpack and a solar

zenith angle of 49.5°, whose cosine value (0.65) represents the insolation-weighted
mean solar zenith cosine in the sunlit Earth hemisphere (Dang et al., 2015). The snow
grain optical effective radius was varied from 50 to 1000 μm (at 50-μm intervals) to
characterize snow aging. Moreover, the BC concentration was assumed to range from
0-1000 ng g$^{-1}$ (at 10-ng g$^{-1}$ intervals) to simulate clear to polluted snow, which was
based on global field observations of the BC concentration in snowpacks mostly below
1000 ng g$^{-1}$ (e.g., Doherty et al., 2010, 2014; Wang et al., 2013; Li et al., 2017, 2018;
Pu et al., 2017; Zhang et al., 2017, 2018). These parameters were also applied in the
subsequent parameterizations (please refer to Section 2.3). In addition, we note that the
SNICAR model adopted in this study is the default version assuming BC-snow external
mixing and spherical snow grains (Flanner et al., 2007). Although the mixing state of
BC and snow grains and the snow grain shape affect the snow albedo, empirical
parameterizations describing the effect of BC internally mixed with snow grains on
snow albedo were developed by He et al. (2018c), and the albedo of a snowpack
consisting of nonspherical snow grains was simulated with smaller spherical grains
(Dang et al. 2016). Therefore, users may combine the empirical parameterizations
developed by He et al. (2018c) and Dang et al. (2016) with our empirical

parameterizations (please refer to Section 2.3) to study the effect of the internal mixing

of BC with snow grains, snow grain shape, and coated BC particles on snow albedo.

Regarding the SNICAR snow albedo simulations considering uncoated BC

particles, the concentrations of both BC and the other particles were directly input.

Regarding coated BC particles, the optical parameters (MEC, SSA, and $g$) of IMPs (as

calculated above) were first archived as lookup tables within the SNICAR model, and

the IMP concentration was then input.

**2.2 Calculation of the broadband snow albedo**

The spectral albedo ($\alpha_\lambda$) was integrated over the solar spectrum ($\lambda = 300\text{–}2500$ nm)

and weighted based on the incoming solar irradiance ($S_\lambda$) to calculate the broadband

snow albedo ($\alpha_{integrated}$):

$$\alpha_{integrated} = \frac{\int \alpha_\lambda S_\lambda d_\lambda}{\int S_\lambda d_\lambda} \tag{1}$$

The considered incoming solar irradiance was the typical surface solar spectrum

for mid–high latitudes from January to May, calculated with the Santa Barbara Discrete

Ordinate Radiative Transfer Atmospheric Radiative Transfer (SBDART) model (Pu et

al., 2019), which is one of the most widely applied models in radiative transfer

simulations (for further details, please refer to Ricchiazzi et al. 1998).

**2.3 Parameterizations**

In the original SNICAR model, the BC coating effect is simply parameterized with

an absorption enhancement factor of ~1.5 (Flanner et al., 2007). However, the BC

coating effect on snow albedo widely varies and depends on the BC concentration,

core/shell ratio, snow grain size, and type of particle shell (please refer to Section 3.3).

In view of this complexity, more explicit parameterizations were developed in this

study:

$$E_{\alpha,\text{integrated}} = \frac{\alpha_{int,\text{integrated}}}{\alpha_{ext,\text{integrated}}} \tag{2}$$

where $\alpha_{ext,\text{integrated}}$ and $\alpha_{int,\text{integrated}}$ are the broadband snow albedos for EMPs

and IMPs, respectively. Following a previous empirical formulation (Hadley and

Kirchstetter, 2012), $E_{\alpha,\text{integrated}}$ was parameterized as:

$$E_{\alpha,\text{integrated,para}} = a_0 \times (C_{BC})^{a_1} + a_2 \tag{3}$$

$$a_1 = b_0 \times (\log_{10}(R_{ef}/50))^{b_1} \tag{4}$$

where $E_{\alpha,\text{integrated,para}}$ is the parameterization of $E_{\alpha,\text{integrated}}$, $C_{BC}$ is the BC

concentration, and $R_{ef}$ is the snow grain radius. The terms $a_0$, $a_1, a_2$, $b_0$, and $b_1$

are empirical coefficients depend on the core/shell ratio and type of particle shell. To

enhance the precision, the above parameterizations were divided into two groups: the

first to consider relatively clean snow (at a BC concentration < 200 ng g$^{-1}$) and the

second to consider relatively polluted snow (200 ng g$^{-1}$ < BC concentration < 1000 ng

g$^{-1}$).

**2.4 Calculation of the in situ snow albedo and radiative forcing**

In situ broadband clear-sky ($\alpha_{\text{integrated}}^{\text{clear,in−situ}}$) and cloudy-sky ($\alpha_{\text{integrated}}^{\text{cloudy,in−situ}}$) albedos

were separately calculated based on corresponding in situ snow-LAP parameters and

SBDART simulated clear- and cloudy-sky incoming solar irradiance levels, respectively. We assumed a semi-infinite snowpack due to the limited available snow depth measurements. The BC and OC concentrations were collected from in situ field measurements (e.g., Doherty et al., 2010, 2014; Wang et al., 2013; Li et al., 2017, 2018; Pu et al., 2017; Zhang et al., 2017, 2018). A snow grain radius of 100 (1000) μm was assumed for fresh (old) snow, which is comparable to previous observations at mid-high latitudes in winter (Wang et al., 2017; Shi et al., 2020). The value of the solar zenith angle was calculated based on the longitude, latitude, and sampling time at each sampling site. The in situ all-sky albedo ($\alpha_{\text{integrated}}^{\text{all-sky,in-situ}}$) was then calculated based on weighted clear- and cloudy-sky albedo values depending on the cloud fraction (CF), given as:

$$\alpha_{\text{integrated}}^{\text{all-sky,in-situ}} = \text{CF} \times \alpha_{\text{integrated}}^{\text{cloudy,in-situ}} + (1 - \text{CF}) \times \alpha_{\text{integrated}}^{\text{clear,in-situ}} \tag{5}$$

The in situ radiative forcing due to LAPs was calculated by multiplying the derived broadband albedo reduction by the downward shortwave flux at the snow surface (Dang et al., 2017). We note that the radiative forcing was calculated with the January-February average solar radiation in NA and NC and the April-May average solar radiation in the Arctic and TP according to the periods of corresponding field campaigns. In this study, we mainly estimated the relative impacts of internal and external mixing on snow albedo and radiative forcing, which are hence not influenced by the chosen solar radiation level. Figure S2 shows spatial distributions of the solar flux and cloud fraction, which were obtained from the Clouds and the Earth's Radiant Energy System

(CERES) (https://ceres.larc.nasa.gov/products.php?product=SYN1deg).
**3 Results and discussion**
**3.1 Impact on particle light absorption**
Figure 1b and 1d show the light absorption enhancement and $E_{abs}$, respectively,
due to coated BC particles. $E_{abs}$ is defined as the ratio of the light absorption due to
coated ($LA_{int}$) and uncoated BC particles ($LA_{ext}$) ($E_{abs} = \frac{LA_{int}}{LA_{ext}}$). Based on Bond et al.
(2006), we show the most common core/shell ratios (the ratio of the diameter of the
whole particle to that of the BC core) of 1.2, 1.5, 2.0, and 2.5 in real environments to
represent the thickness of shells, and we considered detailed core/shell ratios ranging
from 1.1 to 3.0 (at intervals of 0.1) in the parameterizations (see Section 3.5). $E_{abs}$ varies
with the wavelength and increases with the core/shell ratio, in contrast to the default
$E_{abs}$ value employed in the original SNICAR model, which remains constant. Regarding
nonabsorbing shells, the light absorption of IMPs is higher than that of EMPs across all
wavelengths (300–1400 nm). Regarding absorbing shells, $E_{abs}$ is similar to that of
nonabsorbing shells in the NIR range but decreases in the visible (VIS) light and
ultraviolet (UV) light ranges, which indicates that absorbing shells reduce whole-
particle light absorption and negatively contribute to $E_{abs}$. This occurs because
compared to the nonabsorbing shell, the absorbing shell, although it absorbs additional
incident photons, causes fewer photons to reach the core, so that the photons absorbed
by the lensing effect and BC core are reduced. In this case, the number of additional
photons absorbed by the shell is smaller than the number of fewer photons absorbed by
the lensing effect and BC core, causing the total absorption of absorbing shell-coated
BC particles to be lower than that of nonabsorbing shell-coated BC particles (Lack and
Cappa, 2010). Furthermore, the absorbing shell reduces $E_{abs}$ to <1 in the UV range at
high core/shell ratios, suggesting that the lensing effect on absorption at these
wavelengths does not match the BC core absorption reduction, resulting in fewer
photons reaching the core, which is similar to the results reported by Lack and Cappa

7     (2010).

**3.2 Impact on the spectral snow single-scattering properties and albedo**

9        In a real snowpack, BC effectively enhances the snow single-scattering co-albedo

$(1-\omega)$, but its effect on other snow optical parameters, such as the asymmetry factor
and extinction efficiency, is negligible (He et al., 2017). Therefore, we focus our
discussion on the coating-induced enhancement of the snow single-scattering co-albedo
$(E_{1-\omega})$, snow albedo $(E_{\alpha})$, and snow albedo reduction $(E_{\Delta\alpha})$. $E_{1-\omega}$ is defined as the ratio
of the snow single-scattering co-albedo due to coated BC particles $(1-\omega_{int})$ to that due
to uncoated BC particles $(1-\omega_{ext})$ $(E_{1-\omega} = \frac{1-\omega_{int}}{1-\omega_{ext}})$. Similar definitions were adopted for
$E_{\alpha}$ $(E_{\alpha} = \frac{\alpha_{int}}{\alpha_{ext}})$ and $E_{\Delta\alpha}$ $(E_{\Delta\alpha} = \frac{\Delta\alpha_{int}}{\Delta\alpha_{ext}})$, where $\alpha_{int}$ and $\alpha_{ext}$ are the snow albedo values
due to coated and uncoated BC particles, respectively, and $\Delta\alpha_{int}$ and $\Delta\alpha_{ext}$ are the
snow albedo reductions due to coated and uncoated BC particles, respectively.

19       Figure 2 shows the variation in $1-\omega$ and $E_{1-\omega}$ depending on the BC concentration,

core/shell ratio, and coating material. Regarding either the nonabsorbing or absorbing
shell, $1-\omega_{int}$ is usually higher than $1-\omega_{ext}$ in the VIS range, while the coating effect
exerts little impact at wavelengths > 1200 nm because the optical properties of snow
are mainly affected by LAPs in the VIS range but primarily by snow itself at
wavelengths > 1200 nm. In addition, $E_{1-\omega}$ increases with increasing core/shell ratio, and
the wavelength with the maximum $E_{1-\omega}$ value depends on the BC concentration and
core/shell ratio. Moreover, the absorbing shell reveals a negative impact on $E_{1-\omega}$ over
the nonabsorbing shell, especially in the UV range.

7        Snow albedo is notably influenced by various factors, such as the snow grain size,

LAP content, and solar zenith angle, which has been widely examined and verified
through model simulations and experimental measurements in previous studies (e.g.,
Warren and Wiscombe, 1980; Hadley and Kirchstetter, 2012; Wang et al., 2017). In this
study, we mainly focus on the BC coating effect on snow albedo. Figure 3 shows the
spectral snow albedo values due to coated ($\alpha_{int}$) and uncoated BC particles ($\alpha_{ext}$) and
the ratios ($E_\alpha$) of $\alpha_{int}$ to $\alpha_{ext}$. Consistent with $1-\omega$, the impact of the coating effect
on snow albedo is mainly observed at wavelengths < ~1200 nm (Figures 3a versus 3b,
and Figures 3d versus 3e), where the higher the BC concentration is (or the higher the
core/shell ratio is), the larger the difference in snow albedo between uncoated and
coated BC particles. Hadley and Kirchstetter (2012) also found a lower snow albedo
due to internally mixed particles than that due to externally mixed particles. This
phenomenon is also obvious for $E_\alpha$, which decreases with increasing BC concentration
and core/shell ratio in the VIS and NIR ranges (Figure 3c and 3f, respectively). At a
given BC concentration and core/shell ratio, $E_\alpha$ generally decreases with the

wavelength from the UV range to the VIS range and then increases from the VIS range

to the NIR range, which corresponds to the $E_{abs}$ and $E_{1-\omega}$ results. In contrast, the $E_{\alpha}$

values considering the nonabsorbing and absorbing shells are comparable at

wavelengths $> \sim 800$ nm. However, when the wavelength $< \sim 800$ nm, $E_{\alpha}$ considering

the absorbing shell is higher than that considering the nonabsorbing shell, and the

difference increases with decreasing wavelength and increasing core/shell ratio.

Moreover, regarding the absorbing shells, the snow albedo due to coated BC particles

is higher than that due to uncoated BC particles at $< \sim 350$ nm at high core/shell ratios

because the light absorption of internally mixed particles with absorbing shells is lower

than that of externally mixed particles at the same wavelengths, as previously described

in Section 3.1. These results indicate that the material of the particle shell also plays an

important role in snow albedo in the UV and VIS ranges. We note that the solar radiative

flux is very low at wavelengths $< 350$ nm, so that the coating effect at these wavelengths

may contribute little to the total light absorption and broadband snow albedo but may

potentially influence the photochemical reactions in a snowpack (Grannas et al., 2007).

Furthermore, Figure 4 shows the spectral snow albedo reduction caused by coated

$(\Delta\alpha_{int})$ and uncoated BC particles $(\Delta\alpha_{ext})$ and the ratio $(E_{\Delta\alpha})$ of $\Delta\alpha_{int}$ to $\Delta\alpha_{ext}$.

Generally, $\Delta\alpha_{int}$ is larger than $\Delta\alpha_{ext}$, and the core/shell ratio dominates the variation

in $E_{\Delta\alpha}$ across all wavelengths from 300-1400 nm, while the impact of the BC content

is mainly manifested from 500-1000 nm. Consistent with $E_{1-\omega}$ and $E_{\alpha}$, the impact of the

material of the particle shell is negligible at a wavelength $> \sim 800$ nm, but $E_{\Delta\alpha}$ for the

absorbing shell is lower than that for the nonabsorbing shell at a wavelength < ~800
nm. Moreover, $E_{\Delta\alpha}$ is < 1 for the absorbing shell at wavelengths < ~350 nm and high
core/shell ratios. It is noteworthy that the coating effect still yields an obvious impact
on snow albedo reduction at wavelengths > ~1200 nm, which is different from $E_{1-\omega}$ and
$E_{\alpha}$.
**3.3 Impact on the broadband snow single-scattering properties and albedo**
Compared to the spectral optical properties, our broadband results have wider
implications for the research community. Figure 5 shows the spectrally weighted 1-$\omega$
due to coated (1$-\omega_{int, integrated}$) and uncoated BC particles (1$-\omega_{ext, integrated}$) and the ratio
($E_{1-\omega, integrated}$) of 1$-\omega_{int, integrated}$ to 1$-\omega_{ext, integrated}$. In general, 1$-\omega_{int, integrated}$ is larger than
1$-\omega_{ext, integrated}$, and $E_{1-\omega, integrated}$ increases with the BC concentration and core/shell ratio
but is little affected by the snow grain size. $E_{1-\omega, integrated}$ ranges from 1.0 to ~1.35 and
1.0 to ~1.23 for the nonabsorbing and absorbing shells, respectively, with the BC
concentration lower than 1000 ng g$^{-1}$ at core/shell ratios ranging from 1.2-2.5. At a
given BC concentration and core/shell ratio, $E_{1-\omega\ integrated}$ considering the nonabsorbing
shell is higher than that considering the absorbing shell. In addition, $E_{1-\omega, integrated}$
determined with the original SNICAR model, is close to that considering the
nonabsorbing shell at a core/shell ratio of 1.5.
Figure 6 shows the spectrally weighted snow albedo due to coated ($\alpha_{int, integrated}$) and
uncoated BC particles ($\alpha_{ext, integrated}$) and the ratio ($E_{\alpha, integrated}$) of $\alpha_{int, integrated}$ to $\alpha_{ext,}$
$_{integrated}$. Generally, $\alpha_{int, integrated}$ is lower than $\alpha_{ext, integrated}$ by 0 to 0.069 (0 to 0.051), and
$E_{\alpha, \text{integrated}}$ ranges from 1 to ~0.903 (1 to ~0.924) considering the nonabsorbing
(absorbing) shell at BC concentrations from 0 to 1000 ng g$^{-1}$, with the snow grain radius
ranging from 100 to 500 μm, and the core/shell ratio ranging from 1.2 to 2.5. $E_{\alpha, \text{integrated}}$
exhibits a decreasing trend with increasing BC concentration, core/shell ratio and snow
grain size. In addition, the difference between $\alpha_{\text{ext, integrated}}$ and $\alpha_{\text{int, integrated}}$ (or $E_{\alpha, \text{integrated}}$)
for the nonabsorbing shell is larger (or smaller) than that for the absorbing shell. If
considering these coating effects in real environments, e.g., in clean snow, such as in
North America at a typical BC concentration of ~50 ng g$^{-1}$ (Doherty et al., 2014), the
difference between $\alpha_{\text{ext, integrated}}$ and $\alpha_{\text{int, integrated}}$ ranges from 0.002-0.017 and 0.001-
0.012 considering the nonabsorbing and absorbing shells, respectively, at core/shell
ratios from 1.2-2.5 and snow grain radii from 100-500 μm. In contrast, in polluted snow,
such as in northeastern China, the BC concentration is typically ~1000 ng g$^{-1}$ in
industrial regions. The difference between $\alpha_{\text{ext, integrated}}$ and $\alpha_{\text{int, integrated}}$ ranges from
0.008-0.069 and 0.007-0.051 considering the nonabsorbing and absorbing shells,
respectively. These results indicate that the impact of the coating effect on snow albedo
may lead to a reduction in snow albedo by ~2% in clean snow and ~10% in polluted
snow due to coated BC particles below the snow albedo due to uncoated BC particles.
In addition, the sensitivity of $E_{\alpha \text{ integrated}}$ to BC decreases with increasing BC
concentration due to the nonlinear effect of BC on snow albedo (Flanner et al., 2007).
Figure 7 shows the spectrally weighted snow albedo reduction due to coated ($\Delta\alpha_{\text{int,}}$
$_{\text{integrated}}$) and uncoated BC particles ($\Delta\alpha_{\text{ext, integrated}}$) and the ratio ($E_{\Delta\alpha, \text{integrated}}$) of $\Delta\alpha_{\text{int,}}$
$_{integrated}$ to $\Delta\alpha_{ext,\ integrated}$. In contrast to $E_{\alpha,\ integrated}$, $E_{\Delta\alpha,\ integrated}$ is dominated by the
core/shell ratio but slightly depends on the snow grain size (Figures 7c and 6f,
respectively). In addition, $E_{\Delta\alpha,\ integrated}$ exhibits a slight decreasing trend with increasing
BC concentration. Comparing Figure 7c and f, we find that the particle shell material
exerts a distinct integrated impact on $E_{\Delta\alpha}$. $E_{\Delta\alpha,\ integrated}$ mostly ranges from 1.11 to ~1.80
(1.10 to ~1.33) for the nonabsorbing (absorbing) shells at core/shell ratios from 1.2 to
2.5. Our results are comparable to those of a previous study in which the snow albedo
reduction due to BC/snow internal mixing was larger than that due to external mixing
by a factor of 0.2-1.0 (He et al., 2018c). However, the $E_{\Delta\alpha\ integrated}$ value retrieved from
the original SNICAR model demonstrates only a small variation from 1.23–1.31. This
is similar to the nonabsorbing shell at a core/shell ratio of ~1.5, which suggests that the
original SNICAR model only reflects the coating effect on snow albedo reduction at an
intermediate core/shell ratio, which may lead to possible biases ranging from -10% to
50% in snow albedo reduction calculations.
**3.4 Uncertainties**
Although the imaginary RI value of OC has been theoretically calculated (Section
2.1), we note that in a real snowpack, there exists a high uncertainty because the types
and optical properties of OC vary spatially and temporally due to different emission
sources and photochemical reactions in the atmosphere (e.g., Lack and Cappa, 2010).
To address this issue, we tested the degree of influence of the imaginary RI value on
the $E_{\alpha,\ integrated}$, and $E_{\Delta\alpha,\ integrated}$ values by increasing and decreasing the calculated
imaginary RI value by 50% (Figure S1), which studies have revealed to be plausible
(e.g., Lack et al., 2012). We found the imaginary RI uncertainty to be ±1% for $E_{\alpha, integrated}$
and ±5% for $E_{\Delta\alpha, integrated}$.
In addition, observations have demonstrated large variations in the size
distribution of atmospheric and snowpack BC particles (Schwarz et al., 2013), which
may affect the snow optical properties and albedo (He et al., 2018b). Therefore, we
examined the effects of the BC particle size on $E_{\alpha, integrated}$ and $E_{\Delta\alpha, integrated}$ with two
additional BC particle diameters of 50 and 150 nm, which occur within the observed
size ranges (Schwarz et al., 2013) and are comparable to the BC particle sizes adopted
in other studies (e.g., He et al., 2018b). We find that the uncertainty attributed to the BC
particle diameter is ±1% for $E_{\alpha, integrated}$ and ±13% for $E_{\Delta\alpha, integrated}$. According to Equation
2, the uncertainty in $E_{\alpha, integrated}$, is equivalent to that in the snow albedo, and the
uncertainty in $E_{\Delta\alpha, integrated}$, is equivalent to that in the snow albedo reduction. Therefore,
the total uncertainty related to the imaginary RI value and BC particle diameter is ±1.4%
for $E_{\alpha, integrated}$ (snow albedo) and ±13.9% for $E_{\Delta\alpha, integrated}$ (snow albedo reduction).
Another important issue is that in real environments, BC mixtures containing other
species are likely much more complex than are uniform coatings on spheres. Hence, a
core-shell assumption seems somewhat dubious. However, a recent study observing the
individual particle structure and mixing states between glaciers–snowpacks and the
atmosphere (Dong et al., 2018) has found that fresh BC particles are generally
characterized by a fractal morphology, which abundantly occur in the atmosphere. In

contrast, in a snowpack, aged BC particles dominated the BC content, and the mixing

states of aged BC particles largely changed to internal mixing forms with BC at the

core. This process was characterized by the initial transformation from a fractal

structure to a spherical morphology and the subsequent growth of fully compact

particles during the transport and deposition process. Therefore, a core-shell

assumption for coated BC particles in a snowpack seems to be plausible. In addition,

most field measurements have not captured the explicit structure of coated BC particles

due to the limited observation methods (e.g., Doherty et al., 2010, 2014; Wang et al.,

2013; Li et al., 2017, 2018; Pu et al., 2017; Zhang et al., 2017, 2018); therefore, even if

a model of the explicit BC structure was developed, researchers experience difficulties

studying the effect of coated BC particles on snow albedo reduction at present.

Moreover, a core-shell assumption for coated BC particles in the atmosphere has been

widely applied in most global climate models (e.g., Jacobson, 2001; Bond et al. 2013),

so our parameterizations describing coated BC particles in a snowpack may be easily

linked to these models. In summary, we indicate that a core-shell assumption describing

coated BC particles in a snowpack is plausible and practical for field observations and

model simulations at present despite possible uncertainties. However, with the

development of measurement methods and climate models, a more explicit structure

characterizing coated BC particles in a snowpack is actually needed in the future.

**3.5 Parameterizations of the coating effect**

Figure 8 compares parameterized $E_{\alpha, \text{integrated, para}}$ values to SNICAR-modeled $E_{\alpha,}$

$_{integrated}$ values, and Tables S1 and S2, respectively, list the empirical coefficients (please
refer to Section 2.3) derived from the nonlinear regression process. This
parameterization is applicable under the assumptions of a semi-infinite snowpack, BC-
snow external mixing, and spherical snow grains, as mentioned in Section 2. Generally,
$E_{\alpha,\ integrated,\ para}$ and $E_{\alpha,\ integrated}$ exhibit a strong correlation, with $R^2 = 0.988$ (0.986) for
the nonabsorbing shell and $R^2 = 0.987$ (0.986) for the absorbing shell in relatively clean
(polluted) snow, and root mean squared errors of $1.81 \times 10^{-3}$ ($4.70 \times 10^{-3}$) and
$1.41 \times 10^{-3}$ ($3.76 \times 10^{-3}$), respectively. The biases in $E_{\alpha,\ integrated,\ para}$ are the lowest at
intermediate BC concentrations but become relatively high at extremely low or high
concentrations, mainly due to processes within the nonlinear regression method. In
addition, the snow grain size exerts a limited impact on the accuracy of the
parameterized results so that these parameterizations can be applied to either fresh or
old snow types. Overall, the integrated $E_\alpha$ value is suitably reproduced by $E_{\alpha,\ integrated,}$
$_{para}$, and the parameterizations are applicable under various snow pollution conditions
at BC concentrations ranging from 0-1000 ng g$^{-1}$, core/shell ratios ranging from 1.1 to
3.0, and different coating materials (nonabsorbing and absorbing shells). We note that
if the BC concentration is higher than 1000 ng g$^{-1}$, the parameterization describing
relatively polluted snow is also applicable with a low negative bias.
Therefore, future studies may estimate the BC coating effect on snow albedo and
radiative forcing very conveniently by combining the original SNICAR model or other
snow radiative forcing models with our new parameterizations, which may reduce the

snow albedo simulation bias. However, although most global climate models (GCMs) account for coated BC particles in the atmosphere, they barely consider the BC coating effect in snow (Bond et al., 2013). In addition, different GCMs apply varying types of snow radiative transfer models, which indicates that one physical mechanism describing the BC coating effect in snow may not be suitable for all GCMs. Hence, our parameterizations are suitable for climate models and provide an option to capture BC coating effects in snow.

**3.6 Measurement-based estimate of the coating effect**

To evaluate the BC coating effect on both the snow albedo and radiative forcing in a real snowpack, we collected in situ measurements of BC and OC concentrations in snow (Figure 9) during field campaigns in the Arctic in the spring from 2007–2009 (Doherty et al., 2010), in North America from January–March 2013 (Doherty et al., 2014), in northern China from January–February 2010 and 2012 (Ye et al., 2012; Wang et al., 2013), and on the Tibetan Plateau in the spring of 2010 and 2012 (Wang et al., 2013; Li et al., 2017, 2018; Pu et al., 2017; Zhang et al., 2017, 2018). The measurements were separated into four geographical regions (Figure 9c): the Arctic, North America (NA), northern China (NC), and the Tibetan Plateau (TP). An absorbing shell consisting of OC was assumed in the measured snowpack data, which is plausible because previous studies have found that OC is the dominant coating in the atmosphere (e.g., Cappa et al., 2012) and snow (Dong et al., 2018). The OC/BC mass ratio generally ranges from 1 to 10, with the corresponding core/shell ratio ranging from 1.3 to 2.5

(Figure 9b). The average core/shell ratio was the highest (2.45) on the TP, followed by
values of 1.92 and 1.81 in the Arctic and NC, respectively, and was the lowest (1.31) in
NA (Figure 9d). These results reveal that the BC coating effect exerted a larger impact
on snow albedo on the TP than that in the other regions. In this study, the assumption
that all measured OC occurs as a coating on BC particles was mainly adopted to reveal
the upper bound of the coating effect on snow albedo reduction, which is comparable
to previous studies (e.g., He et al. 2018c).
Figure 10 shows statistics of the snow albedo reduction and radiative forcing in
the different regions for fresh snow (snow grain radius =100 μm) and old snow (snow
grain radius =1000 μm). Spatial distributions of the attained snow albedo reduction and
radiative forcing are shown in Figures S3 and S4, respectively. Briefly, the TP
snowpack suffers the highest snow albedo reduction (0.066), and the regional average
snow albedo reduction is lower in NC (0.055), NA (0.009), and the Arctic (0.007) for
fresh snow in the case of external mixing (Figure 10a). Accordingly, the regional
average radiative forcing is 11.63, 4.42, 0.97, and 0.56 W m$^{-2}$ on the TP, NC, NA, and
the Arctic, respectively (Figure 10b). In the case of internal mixing, the regional average
snow albedo reduction is 0.084, 0.065, 0.011, and 0.009 on the TP, NC, NA, and the
Arctic, respectively, with corresponding radiative forcings of 14.84, 5.51, 1.11, and
0.69 W m$^{-2}$, respectively. Figure 11 shows a comparison of internal mixing to external
mixing. In fresh snow, we find that coated BC particles result in greater snow albedo
reductions than those due to uncoated BC particles by factors of 1.27, 1.19, 1.13, and
1.23 on the TP, NC, NA, and the Arctic, respectively (Figures 11a and 11b, respectively).
Correspondingly, we find that the coating effect yields radiative forcing values of 1.27,
1.20, 1.14, and 1.22, respectively, in these regions. The largest (smallest) enhancement
was found on the TP (NA), which corresponds to the highest (lowest) OC/BC mass
ratio and core/shell ratio on the TP (NA). In regard to old snow, the regional average
snow albedo reduction is 0.17 (0.21), 0.14 (0.17), 0.028 (0.033) and 0.022 (0.027) on
the TP, NC, NA, and the Arctic, respectively, for external (internal) mixing (Figure 10c).
The corresponding radiative forcings are 38.2 (47.6), 19.2 (22.7), 4.6 (5.2) and 3.6 (4.6)
W m$^{-2}$, respectively (Figure 10d). The enhancement of snow albedo reduction due to
the BC coating effect is 1.24, 1.15, 1.13, and 1.23 on the TP, NC, NA, and the Arctic,
respectively (Figure 11c). The corresponding radiative forcing reduction is 1.24, 1.16,
1.14, and 1.22, respectively (Figure 11d). The enhancement exhibits a slight decrease
with snowpack aging, which is consistent with the results shown in Figure 7. Notably,
we found that the contribution of the coating effect to light absorption exceeded that of
dust over most areas of northern China after comparison to previous studies of dust in
snow (Wang et al., 2013, 2017; Pu et al., 2017), which further demonstrated the critical
role of the BC coating effect in snow albedo evaluation.

18        In contrast to previous studies, we note that an enhanced light absorption in snow

due to the BC coating effect should be considered, especially in the Arctic and TP.
Arctic sea ice has sharply declined in recent decades (Ding et al., 2019), and climate
models predict a continued decreasing trend (Liu et al., 2020) that is likely to perturb

the Earth system and influence human activities (Meier et al., 2014). Multimodel ensemble simulations have indicated that greenhouse gases cannot fully explain this decline, and recent studies have proposed that BC deposition in snow and sea ice is an important additional contributor (e.g., Ramanathan and Carmichael, 2008). Furthermore, the TP holds the largest ice mass outside the polar regions and acts as a water storage tower for more than 1 billion people in South and East Asia. Tibetan glaciers have rapidly retreated over the last 30 years (Yao et al., 2012), raising the possibility that many glaciers and their freshwater supplies could disappear by the middle of the 21st century. Observed evidence has suggested that BC deposition is a major contributing factor to this retreat (Xu et al., 2009), but the quantitative modeling of the BC effect on glacier dynamics is a challenge, partly because of the incomplete radiative transfer mechanisms within current models. Due to the notable contribution of BC to the retreat in Arctic sea ice and Tibetan glaciers and the strong enhancement of light absorption due to coated BC particles, the coating effect must now be considered in climate models that are designed to accurately reconstruct both historical records and future changes.

**4 Conclusions**

This study evaluated the BC coating effect on snow albedo and radiative forcing by combining the core/shell Mie theory and snow-albedo SNICAR model. We found that the coating effect enhances snow albedo reduction by a factor of 1.11–1.80 for the nonabsorbing shell and 1.10–1.33 for the absorbing shell at BC concentrations below

1000 ng g$^{-1}$, a snow grain radius ranging from 100–500 μm, and a core/shell ratio
ranging from 1.2–2.5. The core/shell ratio plays a dominant role in snow albedo
reduction. Furthermore, the absorbing shell causes a smaller snow albedo reduction
than that caused by the nonabsorbing shell because of the lensing effect, whereby the
absorbing shell reduces photon absorbance in the BC core. Our results effectively
considered the complex enhancement of snow albedo reduction due to the coating effect
in real environments.
Parameterizations describing the coating effect were further developed for
application in snow albedo and climate models. The parameterized and simulated
results exhibit strong correlations in both clean and polluted snowpacks. The root mean
squared error of the parameterized E$_{\alpha, \text{integrated, para}}$ values is small ($1.41 \times 10^{-3}$). A list of
empirical coefficients for parameterizations was provided suitable for most seasonal
snowpack field cases, with BC concentrations lower than 1000 ng g$^{-1}$, snow grain sizes
ranging from 50–1000 μm, and core/shell ratios ranging from 1.1 to 3.0. We
demonstrated that these parameterizations could reduce the simulation bias regarding
local experiments in snow albedo models and, more importantly, could be applied to
GCMs to improve our understanding of how BC in snow affects local hydrological
cycles and the global climate.
Based on a comprehensive set of field measurements across the Northern
Hemisphere, the BC coating effect in real snowpacks was evaluated by assuming the
presence of an absorbing OC shell. The enhancement of snow albedo reduction ranged
from 1.13–1.27, and the enhancement of radiative forcing was 1.14–1.27, which
exceeds the contribution of dust to snow light absorption over most areas of northern
China. Notably, the greatest enhancements were detected on the Tibetan Plateau and in
the Arctic, which may likely contribute to further Arctic sea ice and Tibetan glacier
retreat. Our findings indicated that the coating effect must be considered in future
climate models, in particular to evaluate the climate on the Tibetan Plateau and Arctic
more accurately.

**Conflict of interest**

The authors declare that they have no conflict of interest.

**Acknowledgments**

This research was supported jointly by the National Science Fund for Distinguished Young Scholars (42025102), the National Key R&D Program of China (2019YFA0606801), the National Natural Science Foundation of China (42075061, 41975157 and 41775144), and the China Postdoctoral Science Foundation (2020M673530).

**Author contributions**

X Wang and W Pu invited the project. W Pu and X Wang designed the study. W Pu wrote the paper with contributions from all co-authors. TL Shi processed and analyzed the data.

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

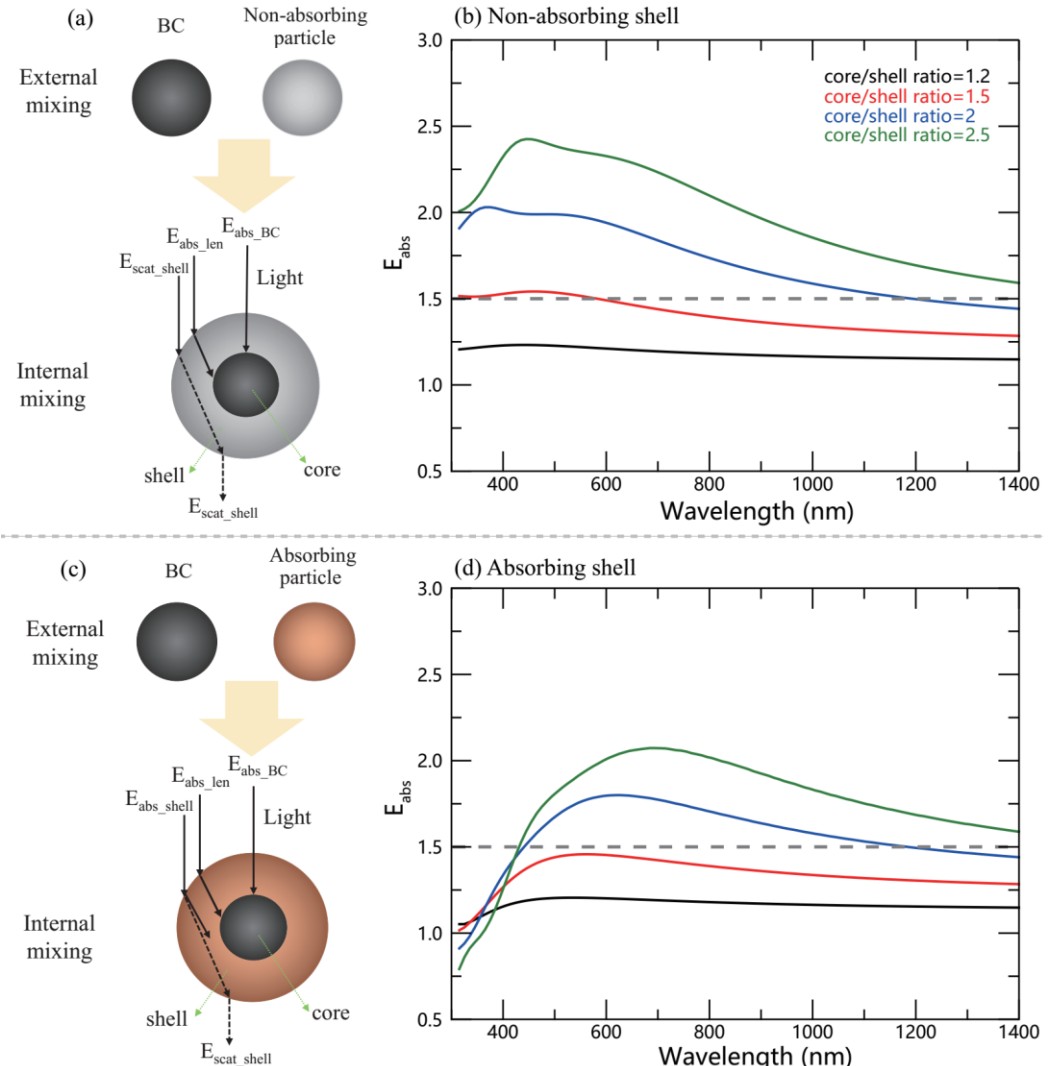

**Figure 1.** Schematic diagrams showing the light absorption of an external mixture and
internal mixture of BC considering (a) a nonabsorbing particle and (c) an absorbing
particle. Additionally, the enhancement of light absorption due to the internal mixture
($E_{abs}$) is compared to that due to the external mixture of BC with (b) nonabsorbing and
(d) absorbing particles. The internal mixed particle was assumed to be a core/shell
structure with a black carbon (BC) core.

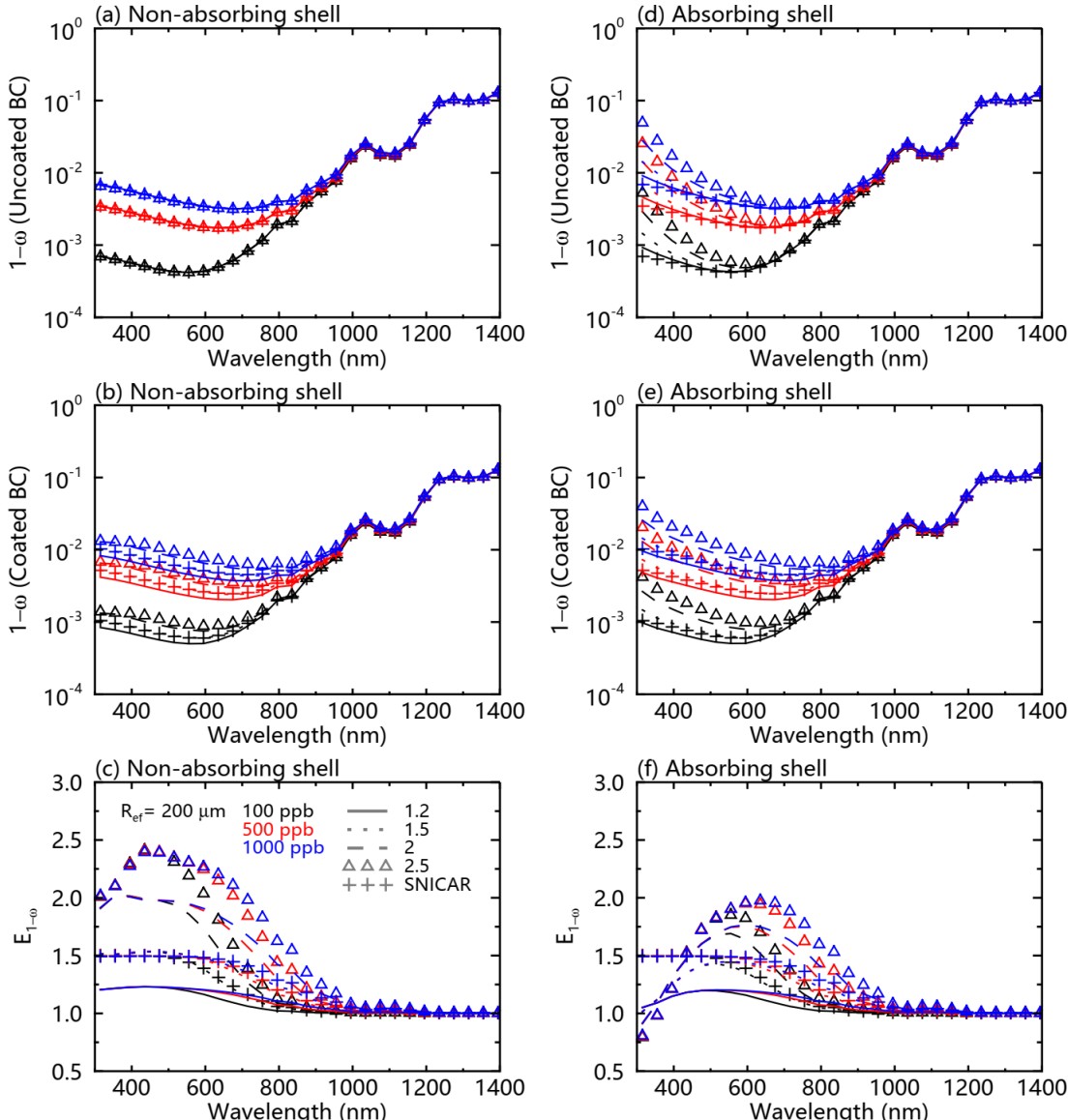

**Figure 2.** Snow single-scattering co-albedo (1–ω) as a function of the wavelength, at different BC concentrations and core/shell ratios for (a) uncoated and (b) coated BC particles under the assumption of a nonabsorbing shell. (d) and (e) are the same as (a) and (b), respectively, but under the assumption of an absorbing shell. (c) shows the ratios of the snow single-scattering co-albedo (E$_{1-ω}$) for coated versus uncoated BC particles under the assumption of a nonabsorbing shell. (f) is the same as (c) but under the assumption of an absorbing shell. The snow grain radius was assumed to be 200 nm.

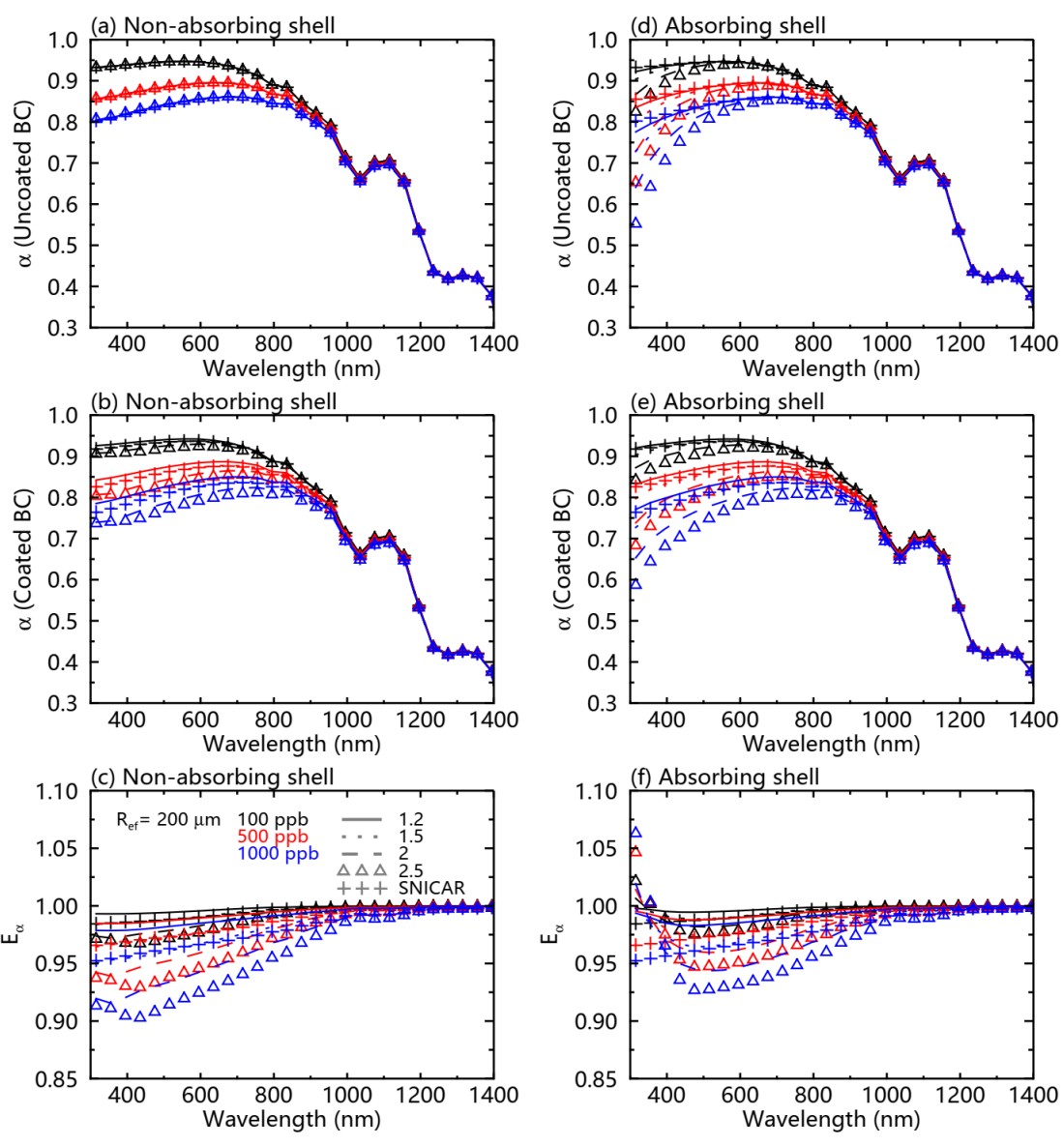

2  **Figure 3.** Same as Figure 2, but for the snow albedo (α).

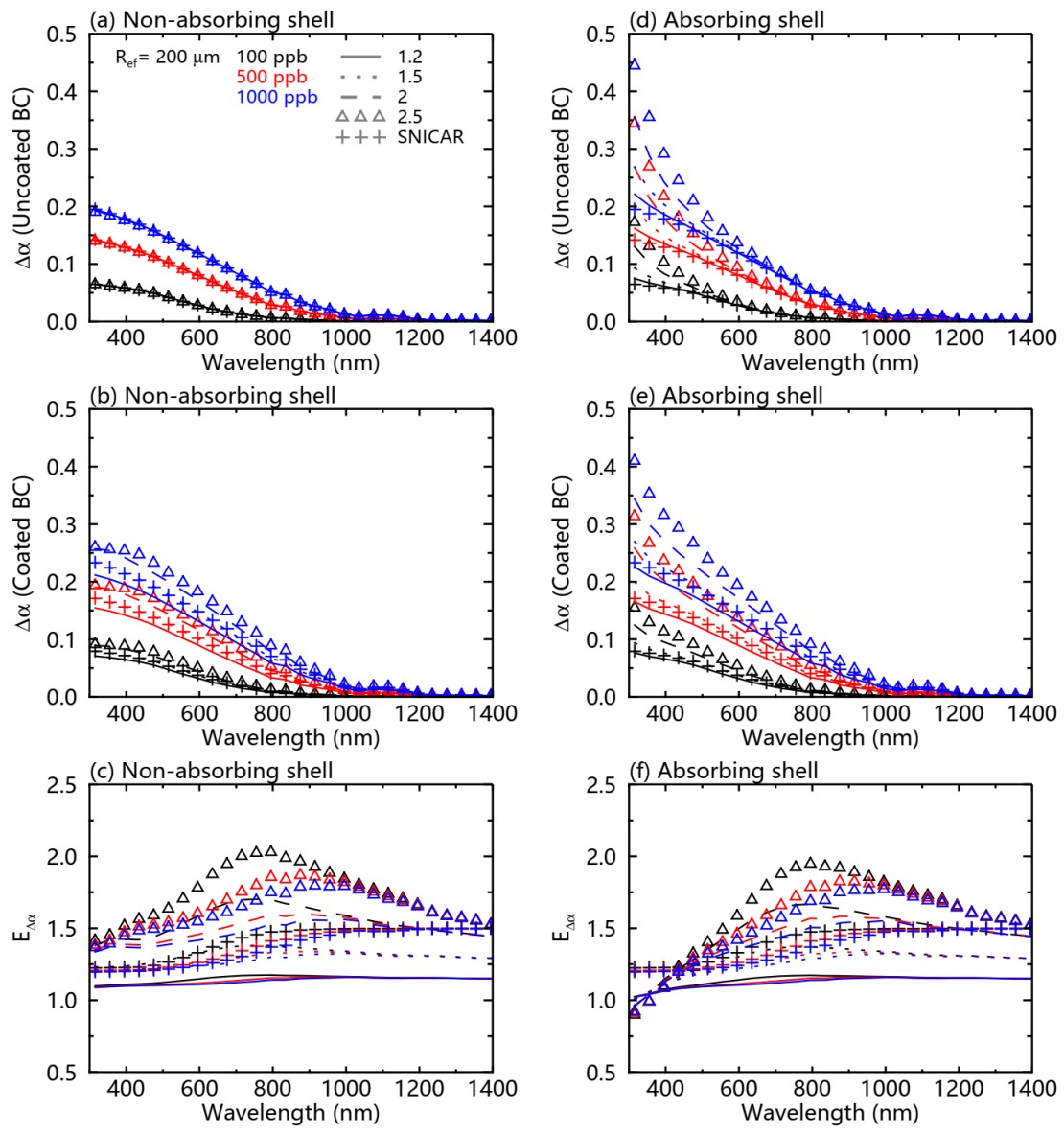

**Figure 4.** Same as Figure 2, but for the snow albedo reduction (Δα).

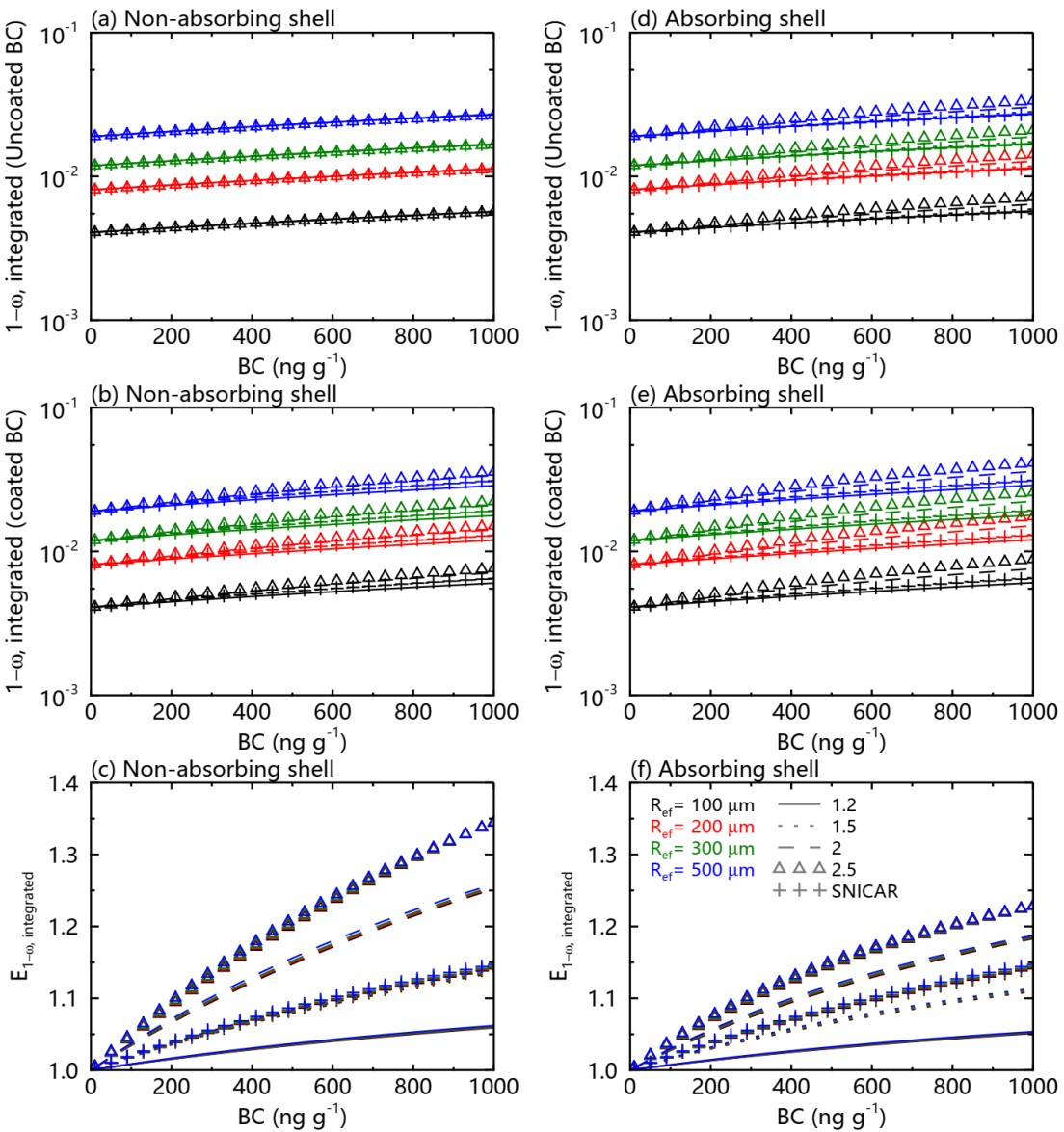

**Figure 5.** Spectrally weighted snow single-scattering co-albedo (1-$\omega_{integrated}$) from 300–2500 nm of the typical surface solar spectrum at mid–high latitudes from January to May for (a) uncoated and (b) coated BC particles under the assumption of a nonabsorbing shell. (d) and (e) are the same as (a) and (b), respectively, but under the assumption of an absorbing shell. (c) shows the ratios ($E_{1-\omega, integrated}$) of the spectrally weighted snow single-scattering co-albedo for coated versus uncoated BC particles under the assumption of a nonabsorbing shell. (f) is the same as (c) but under the assumption of an absorbing shell.

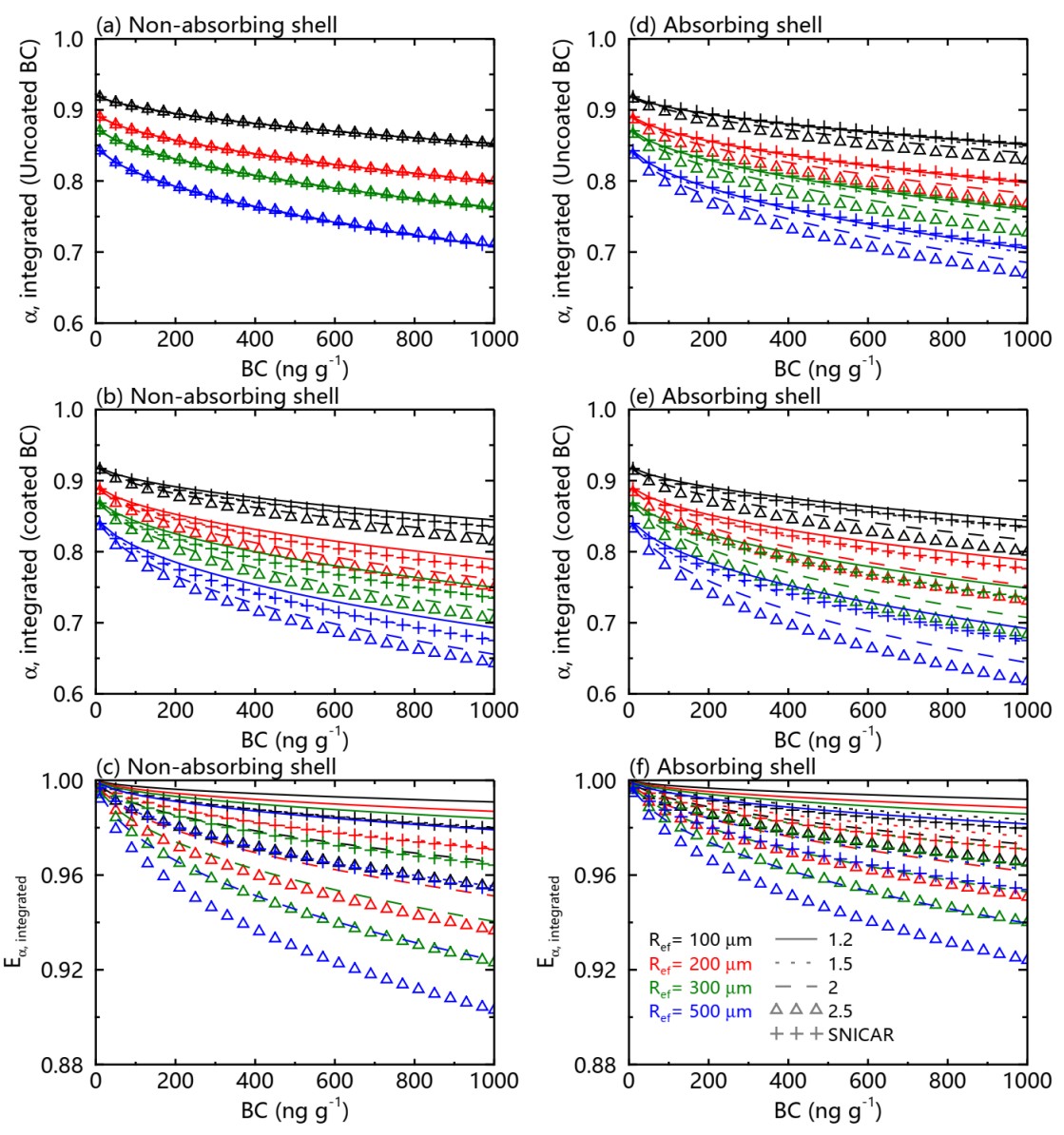

**Figure 6.** Same as Figure 5, but for the snow albedo ($\alpha_{integrated}$).

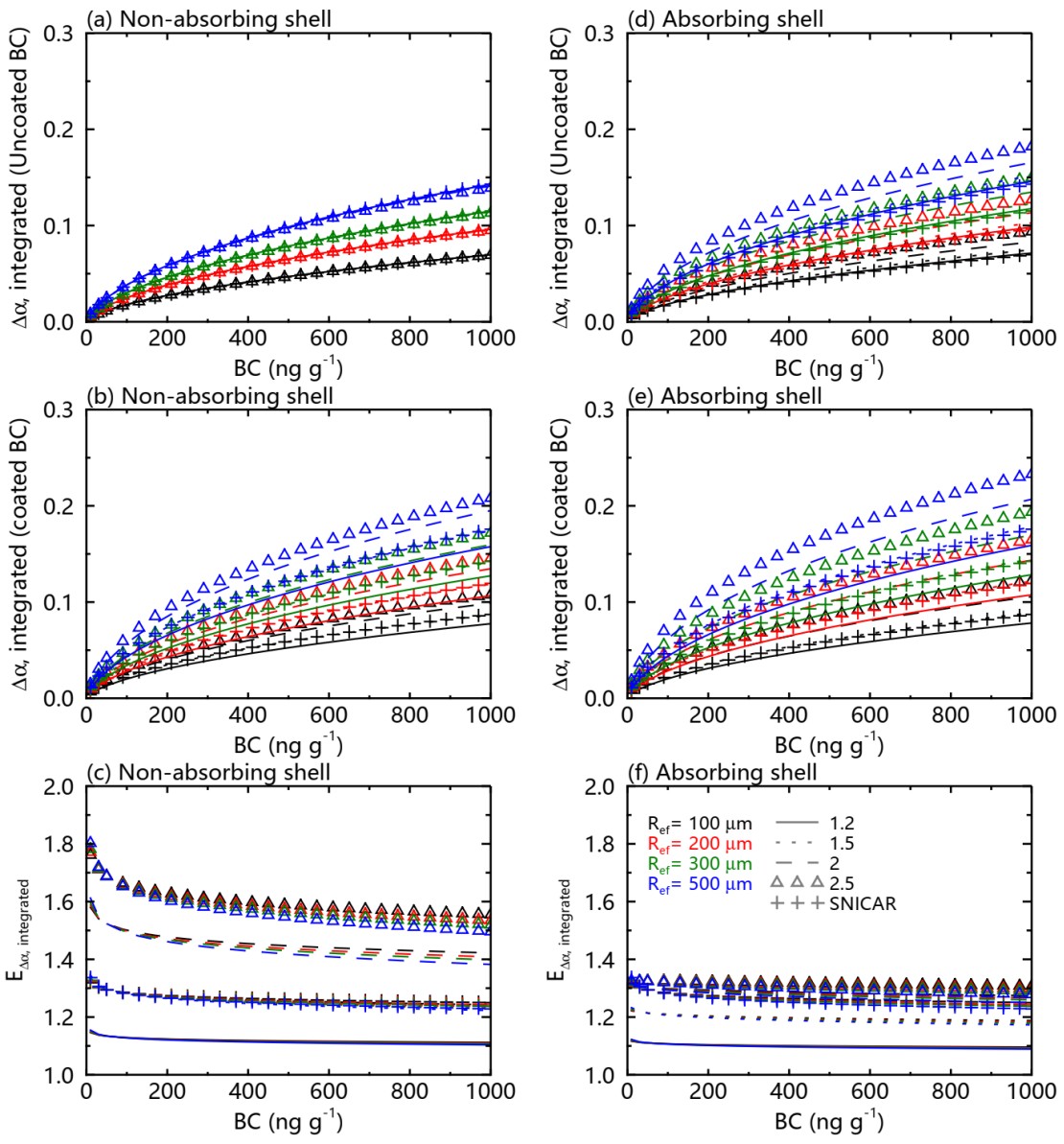

**Figure 7.** Same as Figure 5, but for the snow albedo reduction ($\Delta\alpha_{integrated}$).

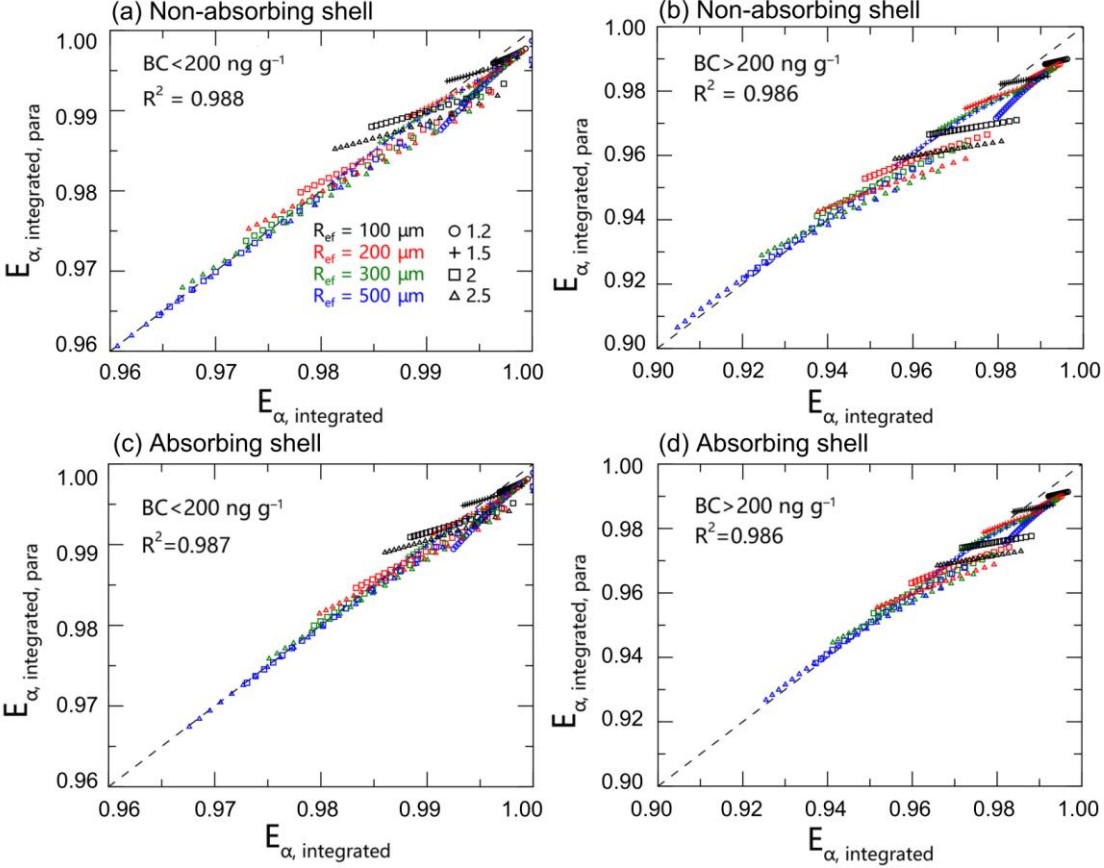

Figure 8. Comparisons of model-calculated $E_{\alpha,\,integrated}$ and parameterized $E_{\alpha,\,integrated,\,para}$ values for (a) relatively clean snow (BC concentration $<200$ ng g$^{-1}$) and (b) relatively polluted snow (BC concentration $>200$ ng g$^{-1}$) for a nonabsorbing shell. (c) and (d) are the same as (a) and (b), respectively, but for an absorbing shell.

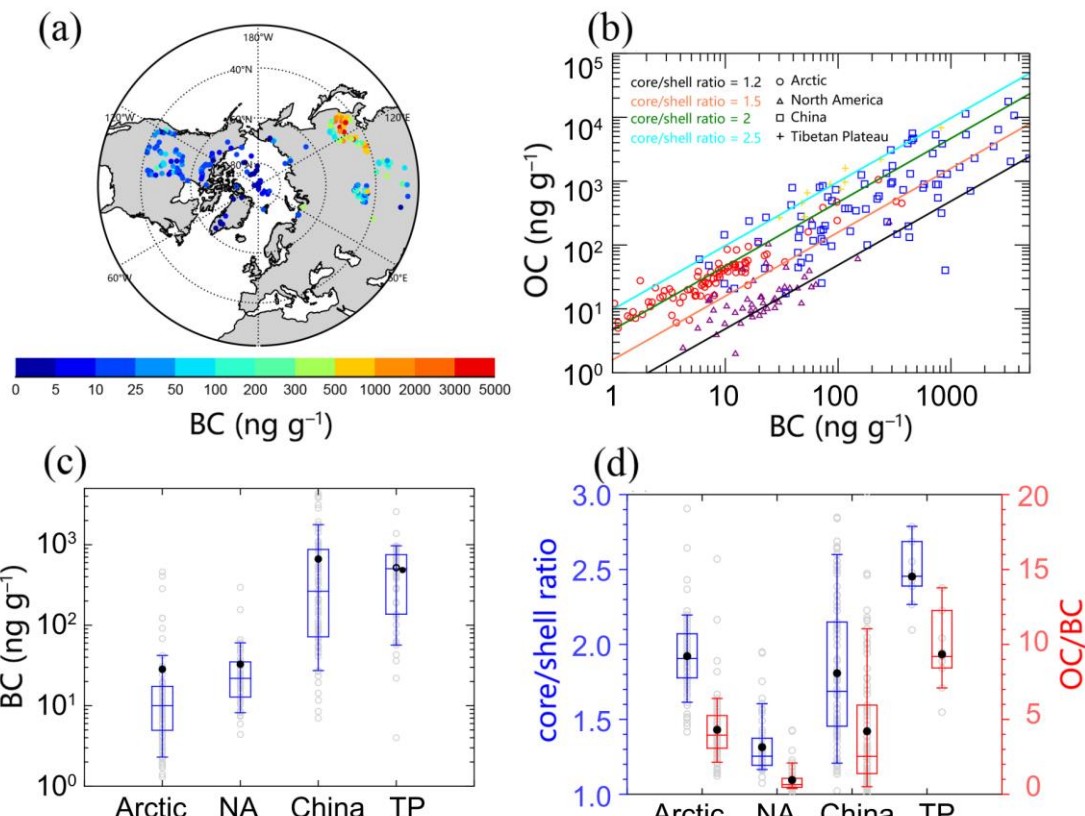

**Figure 9.** (a) Spatial distribution of the measured black carbon (BC) concentration across the Northern Hemisphere. (b) Comparison of the BC and organic carbon (OC) concentrations in the Arctic, North America (NA), northern China (NC) and the Tibetan Plateau (TP). (c) Statistical plots of the BC concentration in the different regions. The boxes denote the 25th and 75th quantiles, the horizontal lines denote the 50th quantiles (median values), the solid dots denote the average values, and the whiskers denote the 10th and 90th quantiles. The in situ data are shown as gray circles. (d) is the same as (c) but for the core/shell ratio and OC/BC mass ratio, assuming a core/shell structure with a BC core and an absorbing OC shell.

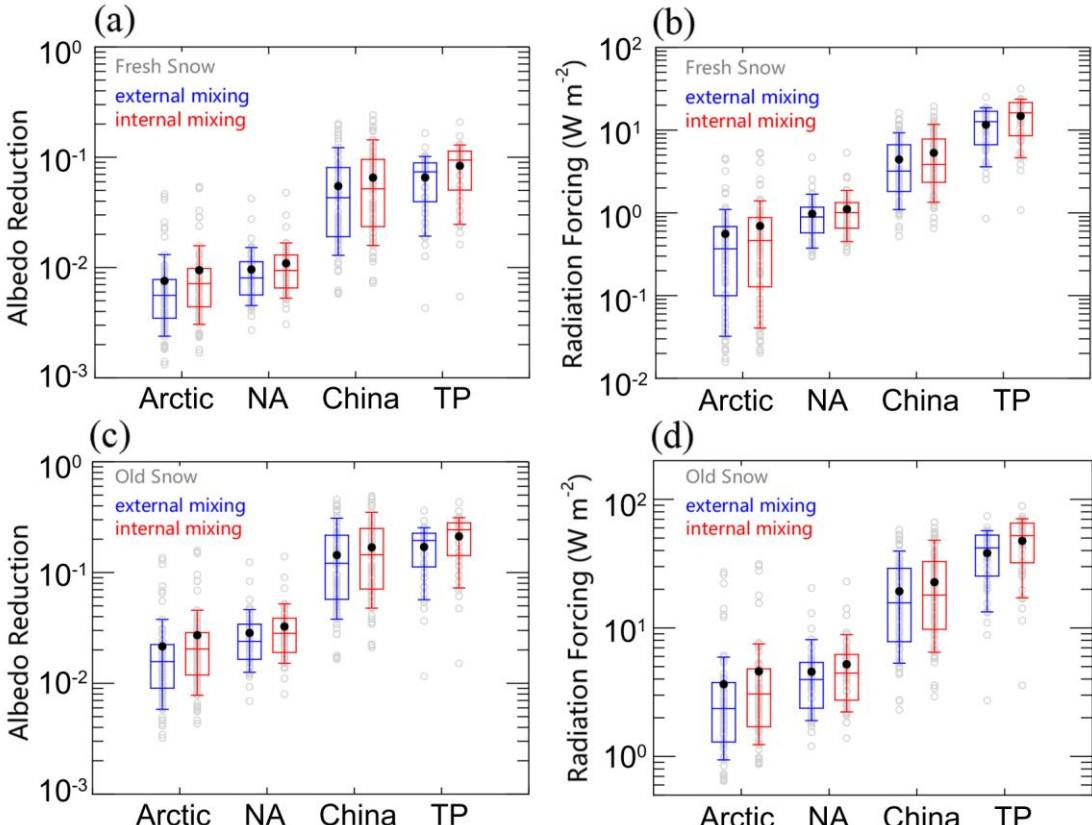

**Figure 10.** Statistical plots of (a) albedo reduction and (b) radiative forcing in the different regions for fresh snow. (c) and (d) are the same as (a) and (b), respectively, but for old snow. The boxes denote the 25th and 75th quantiles, the horizontal lines denote the 50th quantiles (median values), the solid dots denote the average values, and the whiskers denote the 10th and 90th quantiles. The in situ data are shown as gray circles.

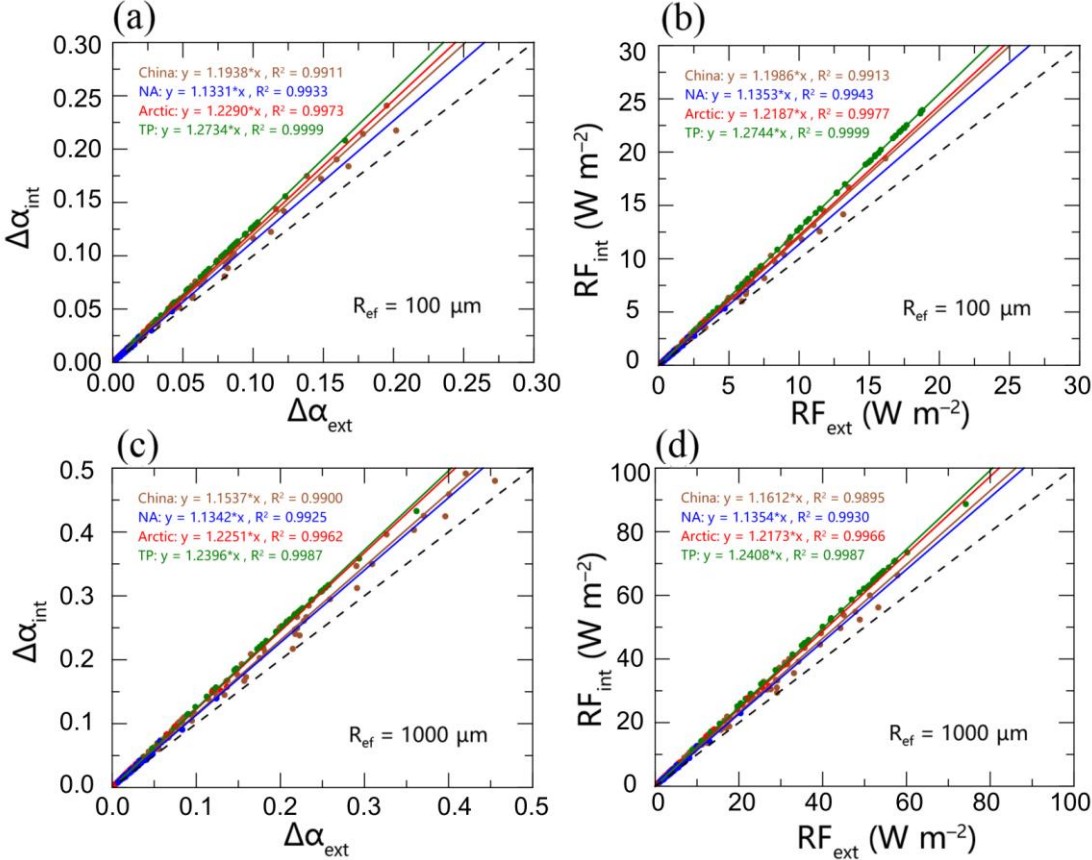

**Figure 11.** Comparisons of (a) snow albedo reduction and (b) radiative forcing via internally mixed particles versus external mixed particles based on in situ measurements of fresh snow (assuming a snow grain radius of 100 µm). (c) and (d) are the same as (a) and (b), respectively, but for old snow and assuming a snow grain radius of 1000 µm.