# Peer review of "Enhancement of snow albedo reduction and radiative forcing"

_The Cryosphere, 2020_

## Referee Comment (RC1) · Anonymous Referee #1 · 24 Oct 2020

The authors investigated the coating effect of BC on BC-induced snow albedo reduction by using core/shell Mie calculations and SNICAR model. They found that BC coating can enhance snow albedo reduction by up to 80% and 30% for non-absorbing and absorbing coating, respectively. They further developed an empirical parameterization for BC coating effect on snow albedo and applied their calculations to different regions based on in-situ measured BC and OC concentrations in snow. This study could help advance our understanding of the role of BC in interacting with snowpack and potentially reduce the uncertainty in estimates of BC-snow albedo radiative effect. The manuscript is generally well-written in terms of language and structure. I have a few comments and suggestions for the authors to consider. Particularly, there are still some places that require more discussions and further clarifications.

[Figure]

Specific comments:

1. The authors assume BC coated by sulfate and OC in snow, which is fine for the purpose of theoretical calculations. However, one important issue related to the coated BC in snowpack is that in reality, many coating materials are soluble (e.g., sulfate and some organics) and will presumably dissolve into BC-containing hydrometeors during wet deposition onto snow surface. Hence, it may not be realistic to assume BC coated by sulfate (and even some OC) in snowpack. I understand this is a complicated problem, and the solubility of BC coating materials heavily relies on the chemical composition. I am not sure if the authors noticed any observations regarding BC coating in snow. If yes, this should be mentioned in the text. If there is no available observation, the authors could at least discuss this issue in the introduction.

2. The authors claimed that "This study is the first to explicitly resolve the optical properties of coated BC in snow ..." in the abstract and main text. However, this is not true. An earlier study (He et al., 2014) has already explicitly resolved the effect of coated BC particles internally and externally mixed with snow grains of different shapes and applied it to the Tibetan Plateau, which is a pioneer study to look at this effect. This earlier study should be briefly discussed in the introduction section and compared with the results from the present study. But it's good to see that the authors here also explored the effect of an absorbing shell.

Reference: He, C., Q. Li, K.-N. Liou, Y. Takano, Y. Gu, L. Qi, Y. Mao, and L. R. Leung (2014), Black carbon radiative forcing over the Tibetan Plateau, Geophys. Res. Lett., 41, 7806–7813, doi:10.1002/2014GL062191.

3. Introduction and Methodology: One important piece that was not mentioned here is the mixing state of BC and snow grains (i.e., internal vs. external) and snow grain shape. Recent studies (e.g., Flanner et al., 2012; Liou et al., 2014; He et al., 2018b) have shown that the BC-snow internal mixing can significantly enhance snow albedo reduction compared with BC-snow external mixing, while nonspherical snow grains

have weaker albedo reduction than snow spheres. This can be briefly discussed in the introduction. Besides, the authors did not mention whether they assumed BC-snow external or internal mixing and whether they assumed spherical snow grains in their SNICAR simulations. By default, SNICAR assumes BC-snow external mixing and snow spheres (Flanner et al., 2007), but a recent study (He et al., 2018c) has extended the SNICAR model to account for BC-snow internal mixing and nonspherical snow grains. So which SNICAR version did the authors use in this study? More details need to be added in the methodology part.

References:

Flanner, M. G., Zender, C. S., Randerson, J. T., and Rasch, P. J.: Present-day climate forcing and response from black carbon in snow, J. Geophys. Res.-Atmos., 112, D11202, https://doi.org/10.1029/2006jd008003, 2007.

Flanner, M. G., Liu, X., Zhou, C., Penner, J. E., and Jiao, C.: Enhanced solar energy absorption by internally-mixed black carbon in snow grains, Atmos. Chem. Phys., 12, 4699–4721, https://doi.org/10.5194/acp-12-4699-2012, 2012.

He, C., Liou, K. N., Takano, Y., Yang, P., Qi, L., and Chen, F.: Impact of grain shape and multiple black carbon inter- nal mixing on snow albedo: Parameterization and radiative effect analysis, J. Geophys. Res.-Atmos., 123, 1253–1268, https://doi.org/10.1002/2017JD027752, 2018b.

He, C., Flanner, M. G., Chen, F, Barlage, M., Liou, K.-N., Kang, S., Ming, J., and Qian, Y.: Black carbon-induced snow albedo reduction over the Tibetan Plateau: uncertainties from snow grain shape and aerosol–snow mixing state based on an updated SNICAR model, Atmos. Chem. Phys., 18, 11507–11527, https://doi.org/10.5194/acp-18-11507-2018, 2018c.

Liou, K. N., Takano, Y., He, C., Yang, P., Leung, R. L., Gu, Y., and Lee, W. L.: Stochastic parameterization for light absorption by internally mixed BC/dust in snow

grains for application to climate models, J. Geophys. Res.-Atmos., 119, 7616–7632, https://doi.org/10.1002/2014JD021665, 2014.

4. Page 6, Line 12: The authors assumed a fixed MAC of 0.3 m2/g at 550 nm for OC. Is there any observation to support this assumption?

5. Page 6, Line 21: The authors seem to assume a fixed monodisperse BC size distribution instead of lognormal distribution, right? Please clarify. Also, what is the assumed shell diameter?

6. Page 9, Lines 4-6: It will be good if the authors can include some comments on how applicable their parameterization is for BC concentration > 1000 ng/g.

7. Page 9, Lines 9-12: The authors assumed an infinite snowpack when applying their calculations to in-situ measurements. Is it because there are no snow depth measurements? Also, what snow-LAP parameters were in-situ measured? Please provide more specifics here.

8. Section 3.1: How did the authors define the variable "Eabs"? A formula will be helpful. Similarly for Section 3.2, definitions of parameters like E_alpha need to be provided in terms of a mathematical expression.

9. Page 11, Line 7: Why does the BC concentration make a negative contribution to E_det(alpha) instead of positive contribution?

10. Page 11, Line 12: Note that the solar radiative flux is very small at wavelengths < 350 nm.

11. Section 3.5: Please include a clarification somewhere in this section to state that this parameterization is under the assumptions of semi-infinite snowpack, BC-snow external mixing, and spherical snow grains.

12. Section 3.6: More descriptions regarding the parameters from observations are needed. For example, did the authors assume semi-infinite snowpack or used measured snow depth in their calculations? Did the authors use time-varying downward solar radiation in the calculation of radiative forcing? How did the authors assume the snow grain size? Is it from observations?

13. It will be good if the authors can include a few sentences to briefly summarize the applicability of their parameterizations in terms of the range of BC concentration, snow condition, snow size, core/shell ratio, etc.

14. Another thing the authors did not mention is the direct and diffuse radiation they assumed in their calculations for both parameterization development and in-situ mea-surement application. If direct radiation was assumed, what was the value of solar zenith angle used? Clarifications need to be added.

---

## Referee Comment (RC2) · Anonymous Referee #2 · 1 Jan 2021

The authors investigated the BC coating effects based on the core/shell Mie theory and the radiative transfer model SNICAR. The paper is generally well-written. It is good for the climate models to consider the BC enhancement on albedo reduction due to the BC coating effects. It is also nice to give options for different core/shell ratios. However, I feel the content is not abundant enough and have some comments below for the authors to consider.

1. In section 3.1, it is not clear to me why the absorptive shell reduce the BC enhancement compared with the non-absorptive shell. And the authors should explain what is the lensing effect. 2. In section 3.2, actually the $E\_1\text{-}w$, $E\_alpha$, and $E\_delta\_alpha$ tell the same story. And the impact of BC coating on spectral characteristics should be consistent with that on broadband characteristics. To make the story full, I suggest

[Figure]

the authors present the direct numbers of snow albedo of various snow cases, e.g. fresh snow, old snow, of different snow depths, with different BC concentrations and core/shell ratios, other than the ratios as E_... 3. In section 3.3, the authors argued that SNICAR consider the BC coating effect of an intermediate core/shell ratio. Well, what is the result of this simplification? 4. It is good for the authors to discuss the uncertainties of imaginary RI values of OC and BC particle sizes in section 3.4. The question is how large the uncertainty is as 1% for E_alpha and 13% for E_delta_alpha? Bias of direct snow albedo is more straightforward. 5. In section 3.5, the authors mentioned the overestimated albedo as around 0.06 in the polluted snow and argued that the new parameterization of BC coating can reduce this overestimation to 0.02. I feel this statement is too strong, as there is huge uncertainty in the measurements of LAPs in snow. It is good for the climate models to have an option for BC coating effects but I suggest not exaggerate this part.

---

## Author Comment (AC1) · 9 Mar 2021

The authors investigated the coating effect of BC on BC-induced snow albedo reduction by using core/shell Mie calculations and SNICAR model. They found that BC coating can enhance snow albedo reduction by up to 80% and 30% for non-absorbing and absorbing coating, respectively. They further developed an empirical parameterization for BC coating effect on snow albedo and applied their calculations to different regions based on in-situ measured BC and OC concentrations in snow. This study could help advance our understanding of the role of BC in interacting with snowpack and potentially reduce the uncertainty in estimates of BC-snow albedo radiative effect. The manuscript is generally well-written in terms of language and structure. I have a few comments and suggestions for the authors to consider. Particularly, there are still some places that require more discussions and further clarifications.

R: We are very grateful for the referee's positive evaluations and valuable comments. The followings are our point by point responses to the comments. Our responses start with "R:".

Specific comments:

1. The authors assume BC coated by sulfate and OC in snow, which is fine for the purpose of theoretical calculations. However, one important issue related to the coated BC in snowpack is that in reality, many coating materials are soluble (e.g., sulfate and some organics) and will presumably dissolve into BC-containing hydrometeors during wet deposition onto snow surface. Hence, it may not be realistic to assume BC coated by sulfate (and even some OC) in snowpack. I understand this is a complicated problem, and the solubility of BC coating materials heavily relies on the chemical composition. I am not sure if the authors noticed any observations regarding BC coating in snow. If yes, this should be mentioned in the text. If there

is no available observation, the authors could at least discuss this issue in the introduction.

R: Thanks for the referee's critical comment. We agreed with the referee's comment that sulfates in the atmosphere acting as coating materials to form coated BC may be partly dissolved during wet deposition. However, this progress doesn't mean that the sulfate-coated BC will be disappeared in real snowpack. A recent study observing individual particle structure and mixing states between the glacier–snowpack and atmosphere based on field measurements and laboratory transmission electron microscope (TEM) and energy dispersive X-ray spectrometer (EDX) instrument analysis (Dong et al., 2018) found that the sulfate components were actually reduced with precipitating snow, while the sulfate-coated BC were still observed during real snowpack in spite of its lower proportion than that in the atmosphere due to dissolution effect. For OC, that study didn't observe reduced OC components in LAPs, which means that dissolution effect will not cause significant reductions of OC-coated BC particles in real snowpack. More notably, that study further found that the proportion of coated BC was even higher in snowpack than that in the atmosphere. All of the above observation results demonstrated that sulfate- and OC-coated are existed in real snowpack and the coated BC particles in snowpack was even more common than that in the atmosphere so that our study focusing on the enhancement of snow albedo reduction and radiative forcing due to coated black carbon in snow is actually helpful in advancing our understanding of the role of BC in interacting with snowpack and potentially reducing the uncertainty in estimates of BC-snow albedo radiative effect.

The referee's comments are still much helpful and make us understand the omission that we didn't mention the facts that coated BC were commonly existed in the snowpack in our study. Therefore, we have added some contents to demonstrate the existence of coated BC in snowpack in "Introduction" section as follows:

*"However, a problem is that whether coated BC is existed in real snowpack because the coating materials (e.g. salts and OC) of coated BC may be dissolved during wet deposition. A recent study observing individual particle structure and mixing states between the glacier–snowpack and atmosphere based on field measurements and laboratory transmission electron microscope (TEM) and energy dispersive X-ray spectrometer (EDX) instrument analysis (Dong et al., 2018) told the truth. They found that the salt-coated BC was still observed in real snowpack in spite of its lower proportion than that in the atmosphere due to the dissolution effect within precipitating snow. For OC, that study didn't observe reduced OC components in LAPs. More notably, that study further found that the proportion of coated BC was even higher in snowpack than that in the atmosphere. All of the above observation results demonstrated that the coated BC particles are existed in real snowpack and even more common than that in the atmosphere. Hence, the climate impacts of BC must be evaluated in the context of the effect of coating on light absorption enhancement."* added from Page 4 Lines 14-21 to Page 5 Lines 1-5

References:

Dong, Z., Kang, S., Qin, D., Shao, Y., Ulbrich, S., and Qin, X.: Variability in individual particle structure and mixing states between the glacier–snowpack and atmosphere in the northeastern Tibetan Plateau, The Cryosphere, 12, 3877-3890, 2018.

2. The authors claimed that "This study is the first to explicitly resolve the optical properties of coated BC in snow . . ." in the abstract and main text. However, this is not true. An earlier study (He et al., 2014) has already explicitly resolved the effect of coated BC particles internally and externally mixed with snow grains of different shapes and applied it to the Tibetan Plateau, which is a pioneer study to look at this effect. This earlier study should be

briefly discussed in the introduction section and compared with the results from the present study. But it's good to see that the authors here also explored the effect of an absorbing shell. Reference: He, C., Q. Li, K.-N. Liou, Y. Takano, Y. Gu, L. Qi, Y. Mao, and L. R. Leung (2014), Black carbon radiative forcing over the Tibetan Plateau, Geophys. Res. Lett., 41, 7806–7813, doi:10.1002/2014GL062191.

R: We indicate that "… the first …" refers to the coated BC by other particles in snowpack, but not the internal mixing of BC and snow grains. But in order to avoid the misunderstanding of the whole research community, the "first" has been removed and this sentence has been revised as "*This study explicitly resolved the optical properties of coated BC in snow ...*". In addition, we have added some discussions about the internally and externally mixing of BC with snow grains and its effect on snow albedo and compared their results with that in our study as follows:

"*Recently, some studies indicated that the mixing state of BC and snow could effectively change snow albedo (Liou et al., 2011, 2014; Flanner et al., 2012; Liu et al., 2012; He et al., 2017, 2018a, b, c). Moreover, snow grain shape also has an important influence on snow albedo (Kokhanovsky and Zege, 2004). Nonspherical snow grains have weaker albedo reduction than snow spheres (He et al., 2018c; Dang et al., 2016*)." added in "Introduction" section from Page 3 Lines 20-21 to Page 4 Lines 1-4

"*Our results were comparable with the previous study that the snow albedo reduction of BC-snow internal mixing is larger than external mixing by a factor of 0.2-1.0 (He et al., 2018c).*"added in Section 3.3 at Page 18 Lines 9-11

References:

Dang, C., Fu, Q., and Warren, S. G.: Effect of Snow Grain Shape on Snow Albedo, J. Atmos. Sci., 73, 3573-3583, 2016.

Flanner, M. G., Liu, X., Zhou, C., Penner, J. E., and Jiao, C.: Enhanced solar energy absorption by internally-mixed black carbon in snow grains, Atmos Chem Phys, 12, 4699-4721, 2012.

He, C. L., Takano, Y., Liou, K. N., Yang, P., Li, Q., Chen, F., He, C., Takano, Y., Liou, K. N., and Yang, P.: Impact of Snow Grain Shape and Black Carbon-Snow Internal Mixing on Snow Optical Properties: Parameterizations for Climate Models, J. Climate, 30, 10019-10036, 2017.

He, C. L., Flanner, M. G., Chen, F., Barlage, M., Liou, K. N., Kang, S. C., Ming, J., and Qian, Y.: Black carbon-induced snow albedo reduction over the Tibetan Plateau: uncertainties from snow grain shape and aerosol-snow mixing state based on an updated SNICAR model, Atmos Chem Phys, 18, 11507-11527, 2018a.

He, C. L., Liou, K. N., and Takano, Y.: Resolving Size Distribution of Black Carbon Internally Mixed With Snow: Impact on Snow Optical Properties and Albedo, Geophys. Res. Lett., 45, 2697-2705, 2018b.

He, C. L., Liou, K. N., Takano, Y., Yang, P., Qi, L., and Chen, F.: Impact of Grain Shape and Multiple Black Carbon Internal Mixing on Snow Albedo: Parameterization and Radiative Effect Analysis, J Geophys Res-Atmos, 123, 1253-1268, 2018c.

Kokhanovsky, A. A., and Zege, E. P.: Scattering optics of snow, Appl Optics, 43, 1589-1602, 2004.

Liou, K. N., Takano, Y., and Yang, P.: Light absorption and scattering by aggregates: Application to black carbon and snow grains, J Quant Spectrosc Ra, 112, 1581-1594, 2011.

Liou, K. N., Takano, Y., He, C., Yang, P., Leung, L. R., Gu, Y., and Lee, W. L.: Stochastic parameterization for light absorption by internally mixed BC/dust in snow grains for application to climate models, J Geophys Res-Atmos, 119, 7616-7632, 2014.

Liu, X., Zhou, C., Penner, J. E., and Jiao, C.: Enhanced solar energy absorption by internally-mixed black carbon in snow grains, Atmos. Chem. Phys., 12, 4699–4721,

https://doi.org/10.5194/acp-12-4699-2012, 2012.

3. Introduction and Methodology: One important piece that was not mentioned here is the mixing state of BC and snow grains (i.e., internal vs. external) and snow grain shape. Recent studies (e.g., Flanner et al., 2012; Liou et al., 2014; He et al., 2018b) have shown that the BC-snow internal mixing can significantly enhance snow albedo reduction compared with BC-snow external mixing, while nonspherical snow grains have weaker albedo reduction than snow spheres. This can be briefly discussed in the introduction. Besides, the authors did not mention whether they assumed BC snow external or internal mixing and whether they assumed spherical snow grains in their SNICAR simulations. By default, SNICAR assumes BC-snow external mixing and snow spheres (Flanner et al., 2007), but a recent study (He et al., 2018c) has extended the SNICAR model to account for BC-snow internal mixing and nonspherical snow grains. So which SNICAR version did the authors use in this study? More details need to be added in the methodology part.

References:

Flanner, M. G., Zender, C. S., Randerson, J. T., and Rasch, P. J.: Present-day climate forcing and response from black carbon in snow, J. Geophys. Res.-Atmos., 112, D11202, https://doi.org/10.1029/2006jd008003, 2007. Flanner, M. G.,

Liu, X., Zhou, C., Penner, J. E., and Jiao, C.: Enhanced solar energy absorption by internally-mixed black carbon in snow grains, Atmos. Chem. Phys., 12, 4699–4721, https://doi.org/10.5194/acp-12-4699-2012, 2012.

He, C., Liou, K. N., Takano, Y., Yang, P., Qi, L., and Chen, F.: Impact of grain shape and multiple black carbon inter- nal mixing on snow albedo: Parameterization and radiative effect analysis, J. Geophys. Res.-Atmos., 123, 1253–1268, https://doi.org/10.1002/2017JD027752, 2018b.

He, C., Flanner, M. G., Chen, F., Barlage, M., Liou, K.-N., Kang, S., Ming, J., and Qian, Y.: Black carbon-induced snow albedo reduction over the Tibetan Plateau: uncertainties from snow grain shape and aerosol–snow mixing state based on an updated SNICAR model, Atmos. Chem. Phys., 18, 11507–11527, https://doi.org/10.5194/acp-18-11507-2018, 2018c.

Liou, K. N., Takano, Y., He, C., Yang, P., Leung, R. L., Gu, Y., and Lee, W. L.: Stochastic parameterization for light absorption by internally mixed BC/dust in snow

R: It is our omission that we didn't discuss the mixing state of BC and snow grains (i.e., internal vs. external) on snow albedo in the "Introduction" section, which has been added as follows:

*"Recently, some studies indicated that the mixing state of BC and snow could effectively change snow albedo (Liou et al., 2011, 2014; Flanner et al., 2012; Liu et al., 2012; He et al., 2017, 2018a, b, c). Moreover, snow grain shape also has an important influence on snow albedo (Kokhanovsky and Zege, 2004). Nonspherical snow grains have weaker albedo reduction than snow spheres (He et al., 2018b; Dang et al., 2016)."* added in "Introduction" section from Page 3 Lines 20-21 to Page 4 Lines 1-4

In this study, we used the default SNICAR that assumes BC-snow external mixing and snow spheres (Flanner et al., 2007), which has been added in "Methods" section. Actually, the mixing state of BC and snow grains, and snow grain shape can affect the snow albedo, which has not been considered in the default SNICAR version. But we note the empirical parameterizations for the effect of BC internally mixed with snow grains on snow albedo has been developed by He et al. (2018c). The albedo of a snowpack consisting of nonspherical snow grains can be mimicked by using a smaller grain of spherical shape Dang et al. (2016). Therefore, users can combine the empirical parameterizations by He et al. (2018c) and Dang et al. (2016) with the empirical parameterizations by us to study the effect of the internal mixing of BC with snow grains, snow grain shape, and coated BC on snow albedo. We have added more discussions in "Methods" section as follows:

*"In addition, we note the SNICAR used in this study was default version that assumes BC-snow external mixing and snow spheres (Flanner et al., 2007). Although the mixing state of BC and snow grains, and snow grain shape can affect the snow albedo, the empirical parameterizations for the effect of BC internally mixed with snow grains on snow albedo has been developed by He et al. (2018c), and the albedo of a snowpack consisting of nonspherical snow grains can be mimicked by using a smaller grain of spherical shape (Dang et al. 2016). Therefore, users can combine the empirical parameterizations by He et al. (2018c) and Dang et al. (2016) with the empirical parameterizations by us to study the effect of the internal mixing of BC with snow grains, snow grain shape, and coated BC on snow albedo."* added at Page 9 Lines 7-17

4. Page 6, Line 12: The authors assumed a fixed MAC of 0.3 $m^2$/g at 550 nm for OC.

Is there any observation to support this assumption?

R: The fixed MAC of 0.3 $m^2$/g at 550 nm for OC was based on the observation from Yang et al. (2009). The reference has been added in the text and the sentence has been revised as *"...Here, a fixed mass absorption coefficient (MAC) for OC of 0.3 $m^2$ $g^{-1}$ at 550 nm, a real RI of 1.55, and a particle diameter of 200 nm were assumed, following the observations of Yang et al. (2009) and the study of Lack and Cappa (2010). The uncertainty of snow albedo of coated BC due to OC MAC will be discussed in Section 3.4."* added at Page 7 Lines 9-13

References:

Lack, D. A., and Cappa, C. D.: Impact of brown and clear carbon on light absorption enhancement, single scatter albedo and absorption wavelength dependence of black carbon, Atmos. Chem. Phys., 10, 4207-4220, 2010.

Yang, M., Howell, S. G., Zhuang, J., and Huebert, B. J.: Attribution of aerosol light absorption

to black carbon, brown carbon, and dust in China - interpretations of atmospheric measurements during EAST-AIRE, Atmos Chem Phys, 9, 2035-2050, 2009.

5. Page 6, Line 21: The authors seem to assume a fixed monodisperse BC size distribution instead of lognormal distribution, right? Please clarify. Also, what is the assumed shell diameter?

R: We have added a clarification that we assumed a fixed monodisperse BC size distribution and the shell diameter of 110 nm–300 nm, and discussed the uncertainty of BC size distribution on our results as

*"...we assumed the BC diameter in snow was 100 nm with a fixed monodisperse size distribution. The uncertainty of snow albedo of coated BC due to fixed BC size distribution will be discussed in Section 3.4. The shell diameter was assumed from 110 nm to 300 nm based on Bond et al. (2006)."*

References:

Bond, T. C., Habib, G., and Bergstrom, R. W.: Limitations in the enhancement of visible light absorption due to mixing state, J. Geophys. Res.-Atmos., 111, D20, 2006.

6. Page 9, Lines 4-6: It will be good if the authors can include some comments on how applicable their parameterization is for BC concentration > 1000 ng/g.

R: Why we chose the BC concentration range of 200-1000 ng g$^{-1}$ for relatively polluted snow is because most of observed BC concentrations in global snowpack are below <1000 ng g$^{-1}$. (e.g. Doherty et al., 2010, 2014; Wang et al., 2013; Li et al., 2017, 2018; Pu et al., 2017; Zhang et al., 2017, 2018). However, we note that if BC concentration is larger than 1000 ng g$^{-1}$, the parameterization for relatively polluted snow is also applicable with a small negative bias based

on the results of Section 3.5. Based on the referee's suggestion, we have added some comments as follows:

*"Meanwhile, BC concentrations were assumed in the range of 0-1000 ng $g^{-1}$ (with a 10-ng $g^{-1}$ interval) to demonstrate clear to polluted snow, which was based on the global field observations with BC concentrations in snowpack mostly below 1000 ng $g^{-1}$ (e.g. Doherty et al., 2010, 2014; Wang et al., 2013; Li et al., 2017, 2018; Pu et al., 2017; Zhang et al., 2017, 2018)."* added in Section 2.1.2 at Page 9 Lines 2-6

*"We note that if BC concentration is larger than 1000 ng $g^{-1}$, the parameterization for relatively polluted snow is also applicable with a small negative bias"* added in Section 3.5 at Page 21 Lines 14-16

References:

Doherty, S. J., Warren, S. G., Grenfell, T. C., Clarke, A. D., and Brandt, R. E.: Light-absorbing impurities in Arctic snow, Atmos. Chem. Phys., 10, 11647-11680, 2010.

Doherty, S. J., Dang, C., Hegg, D. A., Zhang, R., and Warren, S. G.: Black carbon and other light-absorbing particles in snow of central North America, J. Geophys. Res.-Atmos., 119, 12807-12831, 2014.

Li, X., Kang, S., He, X., Qu, B., Tripathee, L., Jing, Z., Paudyal, R., Li, Y., Zhang, Y., and Yan, F.: Light-absorbing impurities accelerate glacier melt in the Central Tibetan Plateau, Sci. Total Environ., 587, 482-490, 2017.

Li, X., Kang, S., Zhang, G., Qu, B., Tripathee, L., Paudyal, R., Jing, Z., Zhang, Y., Yan, F., and Li, G.: Light-absorbing impurities in a southern Tibetan Plateau glacier: Variations and potential impact on snow albedo and radiative forcing, Atmos. Res., 200, 77-87, 2018.

Pu, W., Wang, X., Wei, H., Zhou, Y., Shi, J., Hu, Z., Jin, H., and Chen, Q.: Properties of black carbon and other insoluble light-absorbing particles in seasonal snow of northwestern

China, The Cryosphere, 11, 1213-1233, 2017.

Wang, X., Doherty, S. J., and Huang, J.: Black carbon and other light-absorbing impurities in snow across Northern China, J. Geophys. Res.-Atmos., 118, 1471-1492, 2013.

Zhang, Y., Kang, S., Cong, Z., Schmale, J., Sprenger, M., Li, C., Yang, W., Gao, T., Sillanpää, M., and Li, X.: Light-absorbing impurities enhance glacier albedo reduction in the southeastern Tibetan Plateau, J. Geophys. Res.-Atmos., 122, 6915-6933, 2017.

Zhang, Y., Kang, S., Sprenger, M., Cong, Z., Gao, T., Li, C., Tao, S., Li, X., Zhong, X., and Xu, M.: Black carbon and mineral dust in snow cover on the Tibetan Plateau, The Cryosphere, 12, 413-431, 2018.

7. Page 9, Lines 9-12: The authors assumed an infinite snowpack when applying their calculations to in-situ measurements. Is it because there are no snow depth measurements? Also, what snow-LAP parameters were in-situ measured? Please provide more specifics here.

R: In this study, we assumed an infinite snowpack for all sampling sites due to limited snow depth measurements. Besides, the in-situ measurements applied in SNICAR model included LAPs (i.e. BC and OC) concentrations in snow. The value of solar zenith angle was calculated based on the longitude, latitude, and sampling time at each sampling site. All these specifics and other details have been added in Section 2.4 at Page 11 Lines 13-20.

8. Section 3.1: How did the authors define the variable "Eabs"? A formula will be helpful. Similarly for Section 3.2, definitions of parameters like E_alpha need to be provided in terms of a mathematical expression.

R: We have added a definition of $E_{abs}$ in Section 3.1 at Page 12 Lines 15-17 as that *"Figure 1b and 1d shows the light absorption enhancement, $E_{abs}$ for coated BC. $E_{abs}$ is defined as the ratio of the light absorption for an internal mixture ($LA_{int}$) versus external mixture ($LA_{ext}$) of BC*

$(E_{abs} = \frac{LA_{int}}{LA_{ext}})$. ". Similarly, we have added the definitions of $E_{1-\omega}$, $E_{\alpha}$, and $E_{\Delta\alpha}$ in Section 3.2 at Page 14 Lines 1-5 as that *"The $E_{1-\omega}$ is defined as the ratio of snow single-scattering co-albedo with coated BC ($1-\omega_{int}$) versus that with uncoated BC ($1-\omega_{ext}$) ($E_{1-\omega} = \frac{1-\omega_{int}}{1-\omega_{ext}}$). Similar definitions were used for $E_{\alpha}$ ($E_{\alpha} = \frac{\alpha_{int}}{\alpha_{ext}}$) and $E_{\Delta\alpha}$ ($E_{\Delta\alpha} = \frac{\Delta\alpha_{int}}{\Delta\alpha_{ext}}$), where $\alpha_{int}$ and $\alpha_{ext}$ are snow albedos with coated and uncoated BC, and $\Delta\alpha_{int}$ and $\Delta\alpha_{ext}$ are snow albedo reductions due to coated and uncoated BC."*.

9. Page 11, Line 7: Why does the BC concentration make a negative contribution to E_det(alpha) instead of positive contribution?

R: This is because the nonlinear effect of LAPs such as BC on snow albedo reduction (Flanner et al., 2007), which is that the enhancement capability of snow albedo reduction due to the increase of BC content in high BC concentration condition is lower than that in low BC concentration condition. For example, the ratio of snow albedo reduction by BC of 2*500 ng g$^{-1}$ versus 500 ng g$^{-1}$ is lower than that by 2*100 ng g$^{-1}$ versus 100 ng g$^{-1}$. The coating effect can be equivalent to a increase of BC content so that the BC concentration make a negative contribution to $E_{\Delta\alpha}$ instead of positive contribution.

10. Page 11, Line 12: Note that the solar radiative flux is very small at wavelengths < 350 nm.

R: Yes, the solar radiative flux is very small at wavelengths < 350 nm, but which may influence the photochemical reactions in snowpack. We have added a notice as that *"We note that the solar radiative flux is very small at wavelengths < 350 nm, so that the coating effect at those wavelengths may have little contributions to total light absorption and broadband snow albedos, but which may potentially influence the photochemical reactions in snowpack (Grannas et al., 2007)"* in Section 3.2 at Page 15 Lines 18-21.

References:

Grannas, A. M., Jones, A. E., Dibb, J., Ammann, M., Anastasio, C., Beine, H. J., Bergin, M.,
Bottenheim, J., Boxe, C. S., Carver, G., Chen, G., Crawford, J. H., Domine, F., Frey, M.
M., Guzman, M. I., Heard, D. E., Helmig, D., Hoffmann, M. R., Honrath, R. E., Huey, L.
G., Hutterli, M., Jacobi, H. W., Klan, P., Lefer, B., McConnell, J., Plane, J., Sander, R.,
Savarino, J., Shepson, P. B., Simpson, W. R., Sodeau, J. R., von Glasow, R., Weller, R.,
Wolff, E. W., and Zhu, T.: An overview of snow photochemistry: evidence, mechanisms
and impacts, Atmos Chem Phys, 7, 4329-4373, 2007.

11. Section 3.5: Please include a clarification somewhere in this section to state that this
parameterization is under the assumptions of semi-infinite snowpack, BC-snow external
mixing, and spherical snow grains.

R: A clarification has been added in Section 3.5 at Page 21 Lines 1-3 as that *"This
parameterization is under the assumptions of semi-infinite snowpack, BC-snow external mixing,
and spherical snow grains as mentioned in Section 2"*.

12. Section 3.6: More descriptions regarding the parameters from observations are needed. For
example, did the authors assume semi-infinite snowpack or used measured snow depth in their
calculations? Did the authors use time-varying downward solar radiation in the calculation of
radiative forcing? How did the authors assume the snow grain size? Is it from observations?

For in-situ measurement application, we assumed a semi-infinite snowpack for all sampling
sites due to limited snow depth measurements. While in-situ BC and OC measurements in
snow were used in SNICAR, and snow grain radius of 100 (1000) μm were assumed for fresh
(old) snow, which is comparable to previous observations at mid to high latitudes in winter
(Wang et al., 2017; Shi et al., 2020). The value of solar zenith angle was calculated based on

the longitude, latitude, and sampling time at each sampling site. For the calculation of radiative forcing, we used used the January-February average solar radiation for NA and NC, while April-May average solar radiation for the Arctic and TP according to the periods of filed campaigns. We note that we mainly estimate the relative impact of internal mixing to external mixing on snow albedo and radiative forcing, which is hence not influenced by the chosen solar radiation. All these specifics have been added in Section 2.4 at Page 11 Lines 13-20 and Page 12 Lines 5-10.

References:

Shi, T., Pu, W., Zhou, Y., Cui, J., Zhang, D., and Wang, X.: Albedo of Black Carbon-Contaminated Snow Across Northwestern China and the Validation With Model Simulation, J. Geophys. Res.-Atmos., 125, e2019JD032065, 2020.

Wang, X., Pu, W., Ren, Y., Zhang, X. L., Zhang, X. Y., Shi, J. S., Jin, H. C., Dai, M. K., and Chen, Q. L.: Observations and model simulations of snow albedo reduction in seasonal snow due to insoluble light-absorbing particles during 2014 Chinese survey, Atmos. Chem. Phys., 17, 2279-2296, 2017.

13. It will be good if the authors can include a few sentences to briefly summarize the applicability of their parameterizations in terms of the range of BC concentration, snow condition, snow size, core/shell ratio, etc.

R: Thanks for your suggestions, we have added a brief summary about the applicability of our parameterizations as follow:

*"In addition, the snow grain size has small impacts on the accuracy of parameterized results, so that the parameterizations can be applied in either fresh snow or old snow types. Overall, the $E_{\alpha,\ integrated}$ can be well reproduced by $E_{\alpha,\ integrated,\ para}$ and the parameterizations are*

*applicable in various snow pollution conditions with BC concentrations from 0-1000 ng g$^{-1}$, core/shell ratios from 1.1 to 3.0, and different coating materials (non-absorbing and absorbing shell). We note that if BC concentration is larger than 1000 ng g$^{-1}$, the parameterization for relatively polluted snow is also applicable with a small negative bias."* added at Page 21 Lines 8-16

14. Another thing the authors did not mention is the direct and diffuse radiation they assumed in their calculations for both parameterization development and in-situ measurement application. If direct radiation was assumed, what was the value of solar zenith angle used? Clarifications need to be added.

R: In this study, the direct radiation was assumed for parameterization development with a solar zenith angle of 49.5°, whose cosine value (0.65) represents the insolation-weighted mean solar zenith cosine for sunlit Earth hemisphere (Dang et al., 2015). For in-situ measurement application, the value of solar zenith angle was calculated based on the longitude, latitude, and sampling time at each sampling site. All these specifics have been added in Section 2.1.2 from Page 8 Lines 19-21 to Page 9 Lines 1-7 and Section 2.4 at Page 11 Lines 13-20 and Page 12 Lines 5-10.

---

## Author Comment (AC2) · 9 Mar 2021

The authors investigated the BC coating effects based on the core/shell Mie theory and the radiative transfer model SNICAR. The paper is generally well-written. It is good for the climate models to consider the BC enhancement on albedo reduction due to the BC coating effects. It is also nice to give options for different core/shell ratios. However, I feel the content is not abundant enough and have some comments below for the authors to consider.

R: Thank you very much for the positive evaluations and valuable comments. We have added more contents in the manuscript according to your suggestions and addressed all comments very carefully as detailed below.

1. In section 3.1, it is not clear to me why the absorptive shell reduce the BC enhancement compared with the non-absorptive shell. And the authors should explain what is the lensing effect.

R: When BC is coated with the non-absorptive shell (or the absorptive shell), the light absorption by the BC core can be enhanced, because the shell acts as a lens and focuses more photons onto the core than would reach it otherwise (i.e. lensing effect) (Bond et al., 2006). In addition, the BC light absorption enhancement ($E_{Abs}$) for absorptive shell may differ from that for non-absorptive shell due to either (i) modification of the photon path through the coated particle due to the absorptive shell, thus causing fewer photons to be focused towards the BC core, or (ii) absorption of photons by the absorptive shell, thus causing fewer photons to reach the core. In this case, the total absorption by the coated particle will be conserved (i.e. it does not matter whether a photon is absorbed within the shell or the core), but the magnitude of $E_{Abs}$ has been decreased. When $E_{Abs}>1$, this indicates that photons at that wavelength are still being

focused onto the core due to the lensing effect. However, when $E_{Abs}<1$, this is an indication that the enhancement due to the lensing effect is overwhelmed by absorption by the absorptive shell, similar results have been reported by Lack and Cappa (2010). We have added these discussions in Section 2.1.1 at Page 6 Lines 19-21 and revised the sentences Section 3.1 at Page 13 Lines 6-16.

References:

Bond, T. C., Habib, G., and Bergstrom, R. W.: Limitations in the enhancement of visible light absorption due to mixing state, J Geophys Res-Atmos, 111, 10.1029/2006jd007315, 2006.

Lack, D. A., and Cappa, C. D.: Impact of brown and clear carbon on light absorption enhancement, single scatter albedo and absorption wavelength dependence of black carbon, Atmos Chem Phys, 10, 4207-4220, 10.5194/acp-10-4207-2010, 2010.

2. In section 3.2, actually the E_1-w, E_alpha, and E_delta_alpha tell the same story. And the impact of BC coating on spectral characteristics should be consistent with that on broadband characteristics. To make the story full, I suggest the authors present the direct numbers of snow albedo of various snow cases, e.g. fresh snow, old snow, of different snow depths, with different BC concentrations and core/shell ratios, other than the ratios as E_...

R: The reason why we mainly discussed $E_{1-\omega}$, $E_{\alpha}$, and $E_{\Delta\alpha}$ is snow albedo can be effectively affected by various factors, such as snow grain size, LAP content, solar zenith angle, which has been widely discussed and verified through model simulation and experimental measurements by previous studies (e.g. Hadley and Kirchstetter, 2012; Wang et al., 2017; Warren and Wiscombe, 1980). The use of $E_{1-\omega}$, $E_{\alpha}$, and $E_{\Delta\alpha}$ can make us focus on the impact of BC coating effect on snow albedo, which has been successfully used by previous study (He et al. 2018c). Yet, we agree with the referee's opinions that the direct numbers of snow albedo

of various snow cases are still important for the research community. Hence, according to your suggestions, we have revised the figures, which not only show $E_{1-\omega}$, $E_{\alpha}$, and $E_{\Delta\alpha}$, but also show the direct numbers of snow albedo under various snow cases, e.g. snow grain radius from 100 to 500 µm, BC concentrations from 0 to 1000 ng g$^{-1}$, core/shell ratios from 1.2 to 2.5, and two different coating materials (non-absorbing and absorbing materials):

[Figure]

**Figure 2.** Snow single-scattering co-albedo (1−ω) as a function of wavelength, with different BC concentrations and core/shell ratios for (a) uncoated and (b) coated BC with an assumption of a non-absorbing shell. (d) and (e) are same as (a) and (b), respectively, but with an assumption of an absorbing shell. (c) shows the ratios of snow single-scattering co-albedo ($E_{1−ω}$) for coated versus uncoated BC with an assumption of a non-absorbing shell. (f) is same as (c), but with an assumption of an absorbing shell. The snow grain radius was assumed to be 200 nm.

[Figure]

**Figure 3.** Same as Figure 2, but for snow albedo (α).

[Figure]

**Figure 4.** Same as Figure 2, but for snow albedo reduction ($\Delta\alpha$).

[Figure]

**Figure 5.** The spectrally weighted snow single-scattering co-albedo ($1-\omega_{integrated}$) over 300–2500 nm of a typical surface solar spectrum at mid–high latitude from January to May, for (a) uncoated and (b) coated BC with an assumption of a non-absorbing shell. (d) and (e) are same as (a) and (b), respectively, but with an assumption of an absorbing shell. (c) shows the ratios ($E_{1-\omega,\ integrated}$) of spectrally weighted snow single-scattering co-albedo for coated versus uncoated BC with an assumption of a non-absorbing shell. (f) is same as (c), but with an assumption of an absorbing shell.

[Figure]

**Figure 6.** Same as Figure 5, but for snow albedo ($\alpha_{integrated}$).

[Figure]

**Figure 7.** Same as Figure 5, but for snow albedo reduction ($\Delta\alpha_{integrated}$).

In addition, we have moved Figure S5 to Figure 10 to compare the direct numbers of snow albedo reduction and radiative forcing by coated versus uncoated BC for measurement-based estimate of coating effect in Section 3.6.

[Figure]

**Figure 10.** Statistical plots of (a) albedo reduction, and (b) radiative forcing, in different regions for fresh snow. (c) and (d) Same as (a) and (b), but for old snow. The boxes denote the 25th and 75th quantiles, horizontal lines denote the 50th quantiles (medians), solid dots denote averages, and whiskers denote the 10th and 90th quantiles. In situ data is shown as gray circles.

The detailed contents added in the text can been seen in the revised manuscript in Section 3.2, 3.3 and 3.6.

References:

Hadley, O. L., and Kirchstetter, T. W.: Black-carbon reduction of snow albedo, Nat. Clim. Change, 2, 437-440, 2012.

He, C. L., Liou, K. N., Takano, Y., Yang, P., Qi, L., and Chen, F.: Impact of Grain Shape and Multiple Black Carbon Internal Mixing on Snow Albedo: Parameterization and Radiative

Effect Analysis, J Geophys Res-Atmos, 123, 1253-1268, 2018c.

Wang, X., Pu, W., Ren, Y., Zhang, X. L., Zhang, X. Y., Shi, J. S., Jin, H. C., Dai, M. K., and
Chen, Q. L.: Observations and model simulations of snow albedo reduction in seasonal
snow due to insoluble light-absorbing particles during 2014 Chinese survey, Atmos.
Chem. Phys., 17, 2279-2296, 2017.

Warren, S. G., and Wiscombe, W. J.: A Model for the Spectral Albedo of Snow. 2: Snow
Containing Atmospheric Aerosols, J. Atmos. Sci., 37, 2734-2745, 1980.

3. In section 3.3, the authors argued that SNICAR consider the BC coating effect of an intermediate core/shell ratio. Well, what is the result of this simplification?

R: The result of this simplification that didn't consider the impact of coating materials and core/shell ratio on the BC coating effect only presented a small variation of 1.23–1.31 for $E_{\Delta\alpha, integrated}$. In contrast, our study explicitly resolved the optical properties of coated BC in snow. Our results indicate that a 'BC coating effect' enhances the reduction of snow albedo by a factor of 1.1–1.8 for a non-absorbing shell and 1.1–1.3 for an absorbing shell, depending on BC concentration, snow grain radius, and core/shell ratio. As a result, this simplification may lead to possible biases of -10% to 50% in snow albedo reduction calculation. We have added these discussions in the text at Page 18 Lines 11-16.

4. It is good for the authors to discuss the uncertainties of imaginary RI values of OC and BC particle sizes in section 3.4. The question is how large the uncertainty is as 1% for E_alpha and 13% for E_delta_alpha? Bias of direct snow albedo is more straightforward.

R: According to Equation 2, the uncertainty for $E_{\alpha, integrated}$ is equivalent to that for snow albedo and the uncertainty for $E_{\Delta\alpha, integrated}$ is equivalent to that for snow albedo reduction. As a result, the uncertainties for snow albedo and snow albedo reduction are 1.4% and 13.9%. We have

added this clarification in the text at Page 19 Lines 12-15.

5. In section 3.5, the authors mentioned the overestimated albedo as around 0.06 in the polluted snow and argued that the new parameterization of BC coating can reduce this overestimation to 0.02. I feel this statement is too strong, as there is huge uncertainty in the measurements of LAPs in snow. It is good for the climate models to have an option for BC coating effects but I suggest not exaggerate this part.

R: Thanks for your suggestions. We have removed the sentences about "the overestimated albedo as around 0.06 in the polluted snow and argued that the new parameterization of BC coating can reduce this overestimation to 0.02.". In addition, we have revised the sentences about the application to climate models as follows:

*"On the other hand, although most global climate models (GCMs) account for coated BC in the atmosphere, they barely consider the coating effect for BC in snow (Bond et al., 2013). In addition, different GCMs apply different types of snow radiative transfer models, which means that one physical mechanism responsible for the BC coating effect in snow cannot be suitable for all GCMs. Hence, our parameterizations are good for the climate models to have an option for BC coating effects in snow."* from Page 21 Lines 20-21 to Page 22 Lines 1-5

References:

Bond, T. C., Doherty, S. J., Fahey, D. W., Forster, P. M., Berntsen, T., DeAngelo, B. J., Flanner, M. G., Ghan, S., Karcher, B., Koch, D., Kinne, S., Kondo, Y., Quinn, P. K., Sarofim, M. C., Schultz, M. G., Schulz, M., Venkataraman, C., Zhang, H., Zhang, S., Bellouin, N., Guttikunda, S. K., Hopke, P. K., Jacobson, M. Z., Kaiser, J. W., Klimont, Z., Lohmann, U., Schwarz, J. P., Shindell, D., Storelvmo, T., Warren, S. G., and Zender, C. S.: Bounding the role of black carbon in the climate system: A scientific assessment, J. Geophys. Res.-

Atmos., 118, 5380-5552, 2013.

---

## Author Comment (AC3) · 9 Mar 2021

Comments to the Author:

Dear authors,

Thank you for your submission to TC/TCD. As you may know, papers accepted for TCD appear immediately on the web for comment and review. Before publication in TCD, all papers undergo a rapid access review undertaken by the editor and/or reviewer with the aim of providing initial quality control. It is not a full review and the key concerns are fit to the journal remit, basic quality issues and sufficient significance, originality and/or novelty to warrant publication. As a result, even a manuscript ranked highly during access review can receive a low ranking during full peer review later. Evaluation criteria are found at www.the-cryosphere.net/review/ms_evaluation_criteria.html. Grades are from 1 (excellent) to 4 (poor).

R: Thank the Editor very much for handling the manuscript. We have taken into account all the comments from the Editor and Referees, and made revisions. Please check the responses and the revised manuscript.

ORIGINALITY / NOVELTY (1-4): 2

Although many studies have reported on core-shell Mie calculations of absorption enhancements, few have linked these studies to coated particles in snow.

R: Yes, this study explicitly resolved the optical properties of coated BC in snow, based on core/shell Mie theory and a snow, ice, and aerosol radiative model (SNICAR), which was commonly ignored in previous studies. Our study indicates the nonnegligible enhancement of snow albedo reduction due to the 'BC coating effect'.

SCIENTIFIC QUALITY / RIGOR (1-4): 3

The study seems to adequately explore the relevant parameter space for core-shell Mie calculations, but omits some of the larger picture context of how appropriate these calculations

are for reality. Although Mie calculations are widely used, BC particles are rarely spherical, and BC mixtures with other species are likely much more complex than uniform coatings on spheres. The assumption that all measured OC resides as coating on BC particles also seems somewhat dubious. Furthermore, many of these BC/OC particles reside within ice grains in the snow, as described in previous works, further complicating the radiative transfer in snow.

R: We agree with the Editor that in real environments, BC mixtures with other species are likely much more complex than uniform coatings on spheres. However, a recent study observing individual particle structure and mixing states between the glacier–snowpack and atmosphere based on field measurements and laboratory transmission electron microscope (TEM) and energy dispersive X-ray spectrometer (EDX) instrument analysis (Dong et al., 2018) told the truth. They found that fresh BC particles are generally characterized with fractal morphology, which has a large quantity in the atmosphere. However, in the snowpack, aged BC particles dominated the BC content and the mixing states of aged BC particles change largely to the internal mixing forms with BC as the core. This process is characterized by the initial transformation from a fractal structure to spherical morphology and the subsequent growth of fully compact particles during the transport and deposition process. Therefore, a core-shell assumption for coated BC in snowpack seems to be plausible. In addition, most filed measurements can not capture the explicit structure of coated BC due to limited observation methods, therefore even if a model for explicit BC structure was developed, researchers are hard to use it for studying the effect of coated BC on snow albedo reductions at present. Moreover, a core-shell assumption for coated BC in the atmosphere is widely applied by most global climate models (e.g. Jacobson, 2001; Bond et al. 2013), so that our parameterizations can be easily linked to most climate models. In summary, we indicate that a core-shell assumption for coated BC in snowpack is plausible and practical for field observations and model simulations at present in despite of the possible uncertainties. However, with the

developments of measurement methods and climate models, building a more explicit structure for coated BC in snowpack is actually needed in the future. We have added these discussions in Section 3.4 from Page 19 Lines 15-21 to Page 20 Lines 1-17.

The assumption that all measured OC resides as coating on BC particles were mainly used to show the upper bound of coating effect on snow albedo reduction, which was comparable with the previous studies (e.g. He et al. 2018c). We have added a clarification in the text from Page 22 Line 21 to Page 23 Lines 1-3.

We have added discussions about the mixing of BC and grains in the text, and we demonstrated that users can combine the empirical parameterizations by He et al. (2018c) and Dang et al. (2016) with the empirical parameterizations by us to study the effect of the internal mixing of BC with snow grains, snow grain shape, and coated BC on snow albedo. Details can be seen in our responses to Referee 1 and revised manuscript.

References:

Bond, T. C., Doherty, S. J., Fahey, D. W., Forster, P. M., Berntsen, T., DeAngelo, B. J., Flanner, M. G., Ghan, S., Karcher, B., Koch, D., Kinne, S., Kondo, Y., Quinn, P. K., Sarofim, M. C., Schultz, M. G., Schulz, M., Venkataraman, C., Zhang, H., Zhang, S., Bellouin, N., Guttikunda, S. K., Hopke, P. K., Jacobson, M. Z., Kaiser, J. W., Klimont, Z., Lohmann, U., Schwarz, J. P., Shindell, D., Storelvmo, T., Warren, S. G., and Zender, C. S.: Bounding the role of black carbon in the climate system: A scientific assessment, J. Geophys. Res.-Atmos., 118, 5380-5552, 2013.

Dang, C., Fu, Q., and Warren, S. G.: Effect of Snow Grain Shape on Snow Albedo, J. Atmos. Sci., 73, 3573-3583, 2016.

Dong, Z., Kang, S., Qin, D., Shao, Y., Ulbrich, S., and Qin, X.: Variability in individual particle structure and mixing states between the glacier–snowpack and atmosphere in the

northeastern Tibetan Plateau, The Cryosphere, 12, 3877-3890, 2018.

He, C. L., Liou, K. N., Takano, Y., Yang, P., Qi, L., and Chen, F.: Impact of Grain Shape and Multiple Black Carbon Internal Mixing on Snow Albedo: Parameterization and Radiative Effect Analysis, J Geophys Res-Atmos, 123, 1253-1268, 2018c.

Jacobson, M. Z.: Strong radiative heating due to the mixing state of black carbon in atmospheric aerosols, Nature, 409, 695-697, 2001.

SIGNIFICANCE / IMPACT (1-4): 3

A helpful parameterization is presented that would allow for simplified treatment of coatings, but much uncertainty exists in the applicability of core-shell approximations and of the actual geometry of BC/OC internal mixtures.

R: As seen above, why we used a core-shell assumption for coated BC has been demonstrated, which has been added in the text to improve the reasonability and practicality of our study.

PRESENTATION QUALITY (1-4): 2

The figures seem to present nicely, though the font size of some of the axis labels could be made bigger.

R: We have revised the figures according to your suggestions.

In summary, the paper is a useful and novel contribution and is worth publishing in TCD. Thanks for your contribution.

R: Thanks very much for the Editor's positive evaluations and valuable comments.

---

## Author Response (AR2)

**Editor Decision:**

**Publish subject to technical corrections (22 Mar 2021) by Mark Flanner**

Comments to the Author:

Dear authors -

The reviewers have found your revisions to be suitable and your manuscript ready for publication. Congratulations! I would like to ask that you carefully review your manuscript for grammatical issues to ensure that it is easily readable, and to then upload a final draft of your manuscript. Thanks for your contribution to The Cryosphere!

R: Thank the Editor very much for handling the manuscript. Before we submitted this manuscript to the journal of *The Cryosphere*, it has been improved through scientific English-language editing, as shown by the following certificate.

[Figure]

**Stallard Scientific Editing**
*your trusted partner in English-language excellence*

Richmond Avenue          Tel.:    +64 21 1237099
Nelson 7010                Email:  office@stallardediting.com
New Zealand                Web:    www.stallardediting.com

**Receipt 收据**

| | |
|---|---|
| **Date:** | 8 August 2020 |
| **Job description:** | Scientific English-language editing |
| **Title:** | The primary parameterizations of coated black carbon in snow and its impact on enhanced snow albedo reduction and radiative forcing |
| **Job number:** | 20843 |
| **Client:** | Dr. Wei Pu
Lanzhou University
China |

Nevertheless, we re-edited the language put forth by another language editing company according to your suggestion to make this manuscript more readable as follows.

**INVOICE**

Invoice# SSLFYPXF2

**Balance Due**
CNY0.00

[Figure]

**SPRINGER NATURE**
**Author Services**
**Springer Nature Author Services**
Research Square AJE, LLC
W Main St, Ste 102
Durham, North Carolina, 27701, U.S.A
Tax ID :412141424

| Invoice Date : | 23 Mar 2021 |
| Submission : | SLFYPXF2 |
| Word Count : | 7304 |
| Title : | Enhancement of snow albedo reduction and radiative forcing due to coated black carbon in snow |

Bill To
**Dongyou Wu**
18893121706
Dongyou Wu
lanzhou,GanSu province,china
lanzhou
730107 Gansu
China

| # | Item & Description | Base Price | Amount |
|---|---|---|---|
| 1 | Language Editing: Silver
Language Editing: Silver | 2,825.91 | 2,825.91 |

[revised manuscript text omitted]